# Phospho-regulation, nucleotide binding and ion access control in potassium-chloride cotransporters

Gamma Chi[1,2,†], Rebecca Ebenhoch[1,2,†,‡], Henry Man[1,2,†,§], Haiping Tang[3,†], Laurence E Tremblay[4], Gabriella Reggiano[5], Xingyu Qiu[3], Tina Bohstedt[1,2], Idlir Liko[6], Fernando G Almeida[6], Alexandre P Garneau[4,7], Dong Wang[1,2], Gavin McKinley[1,2], Christophe P Moreau[1,¶], Kiran D Bountra[6], Patrizia Abrusci[1,2,§], Shubhashish M M Mukhopadhyay[1,2], Alejandra Fernandez-Cid[1,2], Samira Slimani[4], Julie L Lavoie[7], Nicola A Burgess-Brown[1,2], Ben Tehan[6], Frank DiMaio[5], Ali Jazayeri[6], Paul Isenring[4], Carol V Robinson[3] & Katharina L Dürr[1,2,6,*] ⓘD

## Abstract

Potassium-coupled chloride transporters (KCCs) play crucial roles in regulating cell volume and intracellular chloride concentration. They are characteristically inhibited under isotonic conditions via phospho-regulatory sites located within the cytoplasmic termini. Decreased inhibitory phosphorylation in response to hypotonic cell swelling stimulates transport activity, and dysfunction of this regulatory process has been associated with various human diseases. Here, we present cryo-EM structures of human KCC3b and KCC1, revealing structural determinants for phospho-regulation in both N- and C-termini. We show that phospho-mimetic KCC3b is arrested in an inward-facing state in which intracellular ion access is blocked by extensive contacts with the N-terminus. In another mutant with increased isotonic transport activity, KCC1Δ19, this interdomain interaction is absent, likely due to a unique phospho-regulatory site in the KCC1 N-terminus. Furthermore, we map additional phosphorylation sites as well as a previously unknown ATP/ADP-binding pocket in the large C-terminal domain and show enhanced thermal stabilization of other CCCs by adenine nucleotides. These findings provide fundamentally new insights into the complex regulation of KCCs and may unlock innovative strategies for drug development.

**Keywords** HDX-MS; nucleotide binding; phospho-regulation; potassium-chloride co-transport; solute carrier

**Subject Categories** Membranes & Trafficking; Structural Biology

**The EMBO Journal (2021) 40: e107294**

See also: **RK Flygaard et al** (July 2021)

## Introduction

Control of intracellular ion homeostasis and cell volume is a fundamental requirement to the normal functioning of all cells (Haas & Forbush, 2000; Russell, 2000; Gamba, 2005; Kahle *et al*, 2015). A family of electroneutral, cation-coupled chloride (CCCs) transporters plays a vital role in this regard (Arroyo *et al*, 2013). CCCs, also known as the SLC12 subfamily of the solute carrier (SLC) superfamily, are secondary active transporters that utilize cellular $Na^+$ or $K^+$ gradients to drive the influx or efflux of $Cl^-$ ions in response to changes in cell volume (Arroyo *et al*, 2013; Gagnon & Delpire, 2013; Flores *et al*, 2019). A hallmark of this transporter family is the highly coordinated, reciprocal regulation of members in two branches of the family. The $Na^+$ transporting importers (NCC, NKCC1-2) facilitate Cl- uptake and are activated by phosphorylation (Rinehart *et al*, 2005; Pieraut *et al*, 2007; Rinehart *et al*, 2009; Gagnon & Delpire, 2010; Melo *et al*, 2013; Lee *et al*, 2014), whereas the $K^+$ transporting effluxers (KCC1-KCC4) are inhibited upon phosphorylation. For all of these transporters, changes in phosphorylation are driven by the same WNK-SPAK/OSR kinases (Kahle *et al*, 2013; de Los Heros *et al*, 2014; Adragna *et al*, 2015; Pisella *et al*, 2019). The major phospho-regulatory threonine residues responsible for activation of NKCCs by

1 Nuffield Department of Medicine, Centre of Medicines Discovery, University of Oxford, Oxford, UK
2 Structural Genomics Consortium, Nuffield Department of Medicine, University of Oxford, Oxford, UK
3 Physical and Theoretical Chemistry Laboratory, University of Oxford, Oxford, UK
4 Department of Medicine, Nephrology Research Group, Faculty of Medicine, Laval University, Quebec City, QC, Canada
5 Department of Biochemistry, University of Washington, Seattle, WA, USA
6 OMass Therapeutics, Ltd., Oxford, UK
7 Cardiometabolic Axis, School of Kinesiology and Physical Activity Sciences, University of Montréal, Montréal, QC, Canada
*Corresponding author (lead contact). Tel: +44 0 1865 548350; E-mail: katharina.duerr@omass.com
†These authors contributed equally to this work (listed in alphabetical order)
‡Present address: MedChem, Boehringer Ingelheim Pharma GmbH & Co. KG, Biberach, Germany
§Present address: Exscientia Ltd, Oxford, UK
¶Present address: Celonic AG, Basel, Germany

SPAK/OSR-1 are located in the N-terminal cytoplasmic domain (Haas *et al*, 1995; Lytle & Forbush, 1996), while two highly conserved phospho-regulatory threonine residues for inhibition by SPAK/OSR-1 are located in the C-terminus of KCC1-4 (Rinehart *et al*, 2009). In addition to these canonical regulatory sites, CCCs have a series of less conserved additional phosphorylation sites in their C- and N-termini, which are controlled by other kinases and likely play a role in fine-tuning of transport activity for each member of the subfamily. There is also a plethora of tissue-specific splice variants with substantial differences in their N-terminal region, including alternative phosphorylation sites.

In the central nervous system (CNS), chloride extrusion by KCCs is not only essential to prevent swelling of neurons (Song *et al*, 2002; Gamba, 2005; Arroyo *et al*, 2013), but also to maintain low intracellular $Cl^-$ concentrations for effective synaptic inhibition by hyperpolarizing Glycine and GABA receptors (Gamba, 2005; Kaila *et al*, 2014). Pathophysiological phenotypes have been reported for a variety of KCC2 and KCC3 mutations (Simon *et al*, 1996; Howard *et al*, 2002; Salin-Cantegrel *et al*, 2007; Gagnon & Delpire, 2013; Kahle *et al*, 2014; Delpire *et al*, 2016; Jin *et al*, 2019) (Appendix Fig S1H). The central importance of fine-tuning KCC3 activity in neuronal cells is highlighted by the finding that peripheral nerve degeneration occurs for both gain-of-function (GOF) and loss-of-function (LOF) mutations in the human *SLC12A6* gene (Flores *et al*, 2019). The recessive Mendelian disorder HMSN/ACC (also known as Andermann syndrome) is caused by a range of different frameshift and premature termination mutations, resulting in complete loss of transport function (Uyanik *et al*, 2006; Akcakaya *et al*, 2018). The recently discovered GOF mutation T991A in KCC3, which results in severe motor neuropathy (Kahle *et al*, 2016), occurs in one of the two canonical phospho-regulatory residues (T991 and T1048 in the A-isoform and T940 and T997 in the B-isoform) that are crucial for phospho-inhibition. KCCs have also been shown to play a number of pathophysiological roles outside of the nervous system.

Although the cryo-EM structure of a full-length *D. rerio* ortholog of NKCC1 has been reported (Chew *et al*, 2019), structural information regarding the human CCC members is currently limited to partial structures of KCC1 and NKCC1 lacking the cytoplasmic domains (Liu *et al*, 2019; Yang *et al*, 2020). While the short N-terminal domain (NTD) is relatively unstructured according to secondary structure predictions, the large C-terminal domain (CTD) exhibits an alternating arrangement of α-helices and β-strands (Appendix Fig S1). The published X-ray structures of the isolated CTDs of a bacterial CCC orthologue from *M. acetivorans* (Warmuth *et al*, 2009) and *C. elegans* KCC1 (Zimanyi *et al*, 2020) provide emerging clues regarding the tertiary structure of this regulatory domain, but only give limited insights into the molecular mechanism of phospho-regulation for human KCCs as the sequence identity with these species is only ~40% in this region (Appendix Fig S2).

Here, we present cryo-EM structures of human KCC1 and KCC3b in non-phosphorylated and phospho-inhibited states, highlighting key features in the architecture of the cytoplasmic N- and C-termini and roles of the phospho-regulatory sites. Via HDX-MS experiments in which a number of KCC1 and KCC3b mutants were characterized, we were able to gain insight into the conformational changes underlying phospho-regulation of KCCs. Furthermore, we discovered an ATP/ADP-binding site in the C-terminal domain and utilize a combination of MD simulations and thermal shift experiments to

characterize the binding properties of different nucleotides for several human CCCs from both $Na^+$ and $K^+$ branches of the family.

# Results

## Design and characterization of phospho-mimetic KCC3b-PM and non-phosphorylated KCC1Δ19 variants

To understand the structural basis of phospho-regulation in potassium-coupled chloride transporters, we used single-particle cryo-electron microscopy to study the full-length wild-type KCC3b transporter and mutants of this transporter in which two canonical threonines (T940 and T997) and a third KCC3-specific phosphorylation site (S45), which was shown to play a role for full activation in response to swelling (Melo *et al*, 2013), were replaced. To gain further insights into the structure KCCs in an activated, non-phosphorylated state, we took advantage of an N-terminally deleted variant of the closely related KCC1 transporter (KCC1Δ19) with deleted N-terminal OSR-1-binding motif H/RFXV (residues 3-6 in full-length KCC1 (Austin *et al*, 2014)). We confirmed lack of phosphorylation at the canonical T926/T983 sites of KCC1Δ19 by proteomics analysis using mass spectrometry (Appendix Fig S6E and F). We utilized this KCC1 mutant as a model system for an activated KCC transporter and determined its structure at 3.12 Å by single-particle cryo-electron microscopy (Fig 1A and Appendix Fig S3). Due to preferred orientation of particles for KCC3, the map resolution for KCC3b-WT was limited to 3.64 Å (Table 1). Unfortunately, the bias towards top views was even worse for mutant KCC3b-PKO and prevented 3D reconstruction beyond 6.5 Å. However, we achieved a map resolution of 3.2 Å for the phospho-inhibited KCC3b-PM variant (Fig 1B and Appendix Fig S4).

Characterization of $Rb^+$ uptake activities in *Xenopus laevis* oocytes revealed a 3- to 5-fold stimulation of $Rb^+$ uptake of the wild-type KCC1 and KCC3 transporters by hypotonic pre-treatment (light grey bars in Fig 1C and D) in comparison to isotonic conditions (dark grey bars in Fig 1C and D). Oocytes expressing mutant KCC1Δ19 exhibited comparable activities under hypertonic and isotonic conditions, which were approximately 65% of the KCC1 WT activity after hypotonic stimulation (light grey bars in Fig 1C). The relative enhancement of transport activity of this construct under isotonic conditions (dark grey bars in Fig 1C) is in line with a loss of phospho-inhibition due to the deleted OSR-1-binding motif. In transport measurements with KCC3b, introducing point mutations S45A/T940A/T997A (KCC3b-PKO) caused substantially higher $Rb^+$ uptake activities under isotonic and hypotonic conditions, resulting in approximately 10-fold enhanced uptake compared to KCC3b-WT under isotonic conditions (dark grey bars in Fig 1D). This phenotype of KCC3-PKO agrees well with a previous study (Melo *et al*, 2013). The corresponding phospho-mimetic mutant S45D/T940D/T997D (KCC3b-PM), on the other hand, showed low $Rb^+$ uptake under both tested conditions, and the approximately 6-fold reduced activity compared to KCC3b-WT after hypotonic stimulation (light grey bars in Fig 1D) confirms that the mutated transporters can no longer be activated by swelling.

## Overall structure and domain arrangement in KCC3b-PM and KCC1Δ19

The overall organization of KCC3b-PM and KCC1Δ19 is similar and resembles the previously solved zebrafish NKCC1 structure (Yang

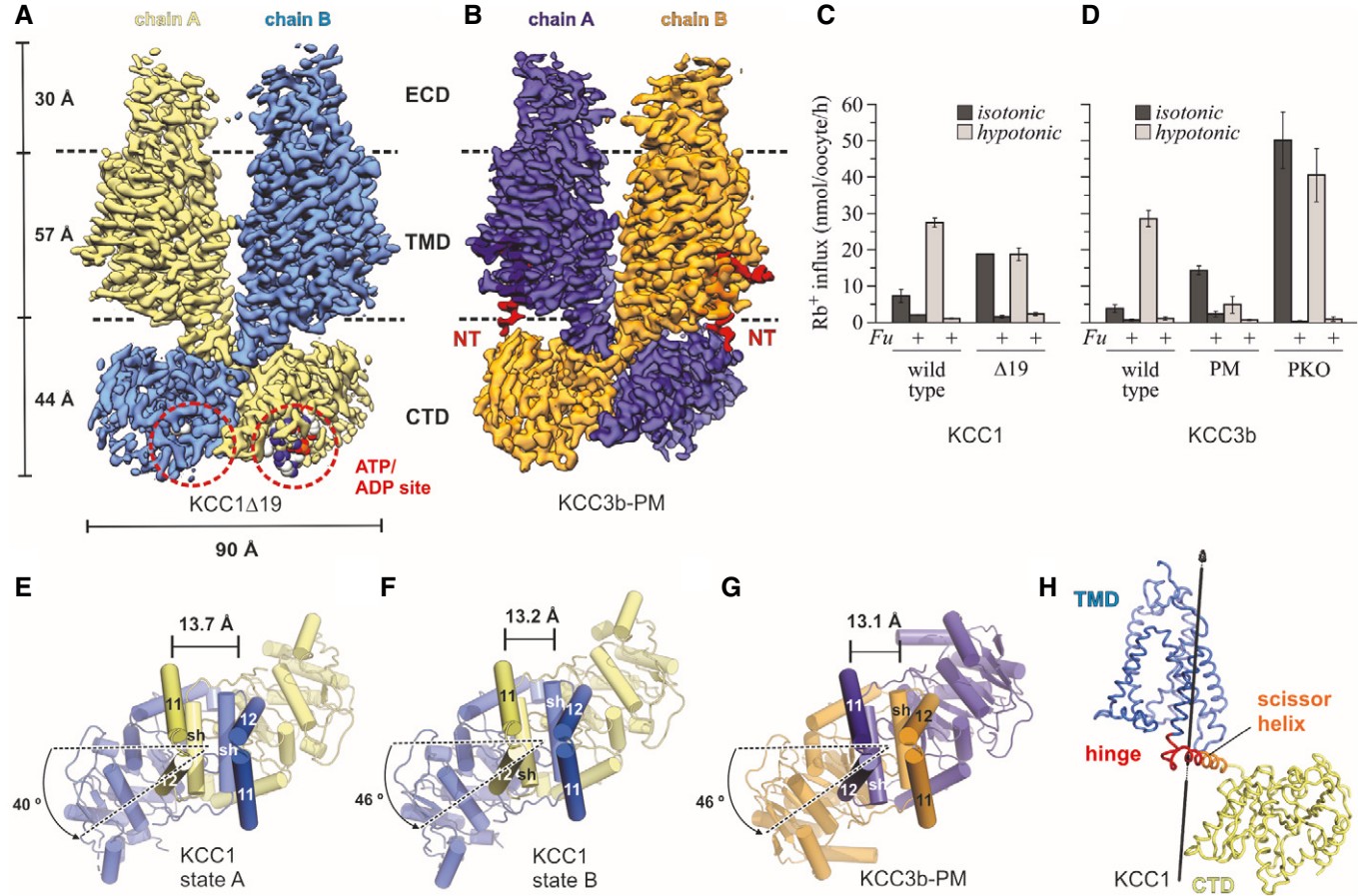

**Figure 1. Overall structure and functional characterization of KCC3b-PM and KCC1Δ19.**

A    Density map of KCC1Δ19, coloured by protomer. ATP bound to the CTD is represented as spheres.
B    Density map of KCC3b-PM, coloured by protomer, with N-terminal region highlighted in bright red.
C, D  Rb$^+$ uptake activity of WT and mutant constructs for KCC1 (C) and KCC3b (D) under isotonic (dark grey bars) and hypotonic conditions (light grey bars). Values are mean (± SE) background-subtracted transport rates of 10 oocytes from 3 to 6 experiments in the presence (indicated by +*Fu*) or absence of 1.5 mM furosemide.
E–G  CTD arrangement of KCC1 (construct Δ19) (E, F) and KCC3b (construct PM: S45D/T940D/T997D, panel G). Of the TMD, only TM helices 11 and 12 are shown for clarity. Cα distances (in Å) between marker atoms S634 (KCC1) and S648 (KCC3b) are indicated by a black line.
H    Hinge region (red ribbon representation) for conformational change between state A and B of KCC1 predicted by DynDom (Lee *et al*, 2003).

*et al*, 2020), showing a typical three-layer architecture (Fig 1A and B), composed of an extracellular domain (ECD), transmembrane domain (TMD) and a large C-terminal domain (CTD), which is organized as a bi-lobed, 10-stranded β-sheet. KCC1Δ19 and KCC3b-PM are domain-swapped dimers with crossover between the two subunits occurring at the scissor helices connecting TMD and CTD. In both structures, TM7 and the cytoplasmic halves of broken helices TM1a and TM6b of the core domain (coloured in orange for KCC3b in Fig EV1A inset) are peeled away from the scaffold domains (TM helices 3–5, 8–10 in wheat in Fig EV1A), locking the TMD monomers into inward-open conformations (Fig EV1B and C). Both structures exhibit dimer interfaces in the transmembrane and C-terminal domains which resemble the dimer arrangement in *Dr*NKCC1 (Fig EV1D and E), but are profoundly different from the reported KCC1 structure without CTD (Fig EV1H and I) and the monomeric KCC4 structure (Reid *et al*, 2020).

Another key feature of KCCs not revealed by the partial KCC structures is the relative orientation of the CTD dimer with respect to TMD. A top view comparison of KCC1Δ19 and KCC3b-PM (Fig 1E–G) illustrates that the CTDs are twisted away from the central two-fold axes of symmetry in a counter-clockwise direction, whereas the *Dr*NKCC1 structure showed a clockwise twist between CTD and TMD (Fig EV1G). Interestingly, 3D classification of the KCC1Δ19 dataset yielded two conformations that differ mainly in the degree of the counter-clockwise CTD twist and the inter-subunit distances between the central TM11/12 helices (Fig 1E and F). The coexistence of two alternating CTD conformations in KCC1Δ19 is indicative of an increased flexibility around the TM-CTD-connecting scissor helices compared to KCC3b-PM. In line with this interpretation, we identified a bending region at residues 655–671 of KCC1 predicted by comparing the structures of state A and B (Fig 1H). Due to this observation, we investigated the flexibility of KCC1 and KCC3b in more depth, which is detailed later in this article.

**Table 1. Cryo-EM data collection, refinement and validation statistics for datasets of KCC3b-PM, KCC3b-WT and KCC1Δ19 constructs under different sample conditions.**

| | KCC3b-PM (Digitonin, NaCl) | | KCC3b-PM (Digitonin, KCl) | KCC3b-WT (LMNG/CHS, NaCl) |
|---|---|---|---|---|
| Construct | KCC3b (M1-S1099, S45D, T940D, T997D) | | KCC3b (M1-S1099, S45D, T940D, T997D) | KCC3b wild type (M1-S1099) |
| **Data collection** | | | | |
| Microscope | Titan Krios (eBIC, UK) | | Titan Krios (MRCEF, UK) | Titan Krios, (ESRF, France) |
| Detector | K3 | | K3 | K2 Summit |
| Voltage (kV) | 300 | | 300 | 300 |
| Magnification | 105,000 | | 105,000 | 130,000 |
| Collection mode | Counting (superresolution) | | Counting (superresolution) | Counting |
| Electron exposure (e/Å$^2$) | 40 | | 40 | 37.8 |
| Number of frames | 45 | | 45 | 40 |
| Pixel size | 0.3255 | | 0.415 | 1.067 |
| Defocus range (μm; steps) | −0.8 to −2.3 (0.3) | | −0.8 to −2.3 (0.3) | −1.4 to −2.8 (0.2) |
| Number of movies | 16,472 | | 4,731 | 3,155 |
| Phase plate used | No | | No | No |
| **Data processing** | **Reference** | **Subclass 1** | | |
| Initial Number of particles | 4,053,596 | | 1,471,447 | 1,582,396 |
| Number of particles into 3D classification | 617,631 | | 1,073,071 | 175,990 |
| Symmetry | C2 | C1 | C1 | C1 |
| Number of particles used for 3D refinement | 408,039 | 302,166 | 293,233 | 175,990 |
| Map resolution (Å; FSC threshold = 0.143) | 3.2 | 3.3 | 4.1 | 3.6 |
| Resolution range (Å) | 2.8–9.2 | 3.0–9.1 | 3.9–8.0 | 3.16–12.7 A |
| Map sharpening B-factor (Å$^2$) | −151 | −122 | −213.78 | −116 |
| **Refinement** | | | | |
| Model resolution (Å; FSC threshold = 0.5) | 3.4 | 3.7 | 4.1 | 4.0 |
| **Model composition** | | | | |
| Non-hydrogen atoms | 13904 | 13915 | 13,393 | 13,144 |
| Protein residues | 1808 | 1819 | 1,758 | 1718 |
| Ligands | 8 | 8 | 4 | 8 |
| **R.M.S.D** | | | | |
| Bond lengths (Å) | 0.010 | 0.006 | 0.011 | 0.010 |
| Bond angles (°) | 0.894 | 0.768 | 0.989 | 1.061 |
| **Validation** | | | | |
| MolProbity score | 1.67 | 1.66 | 2 | 1.75 |
| Clash score | 9.53 | 9.14 | 15.51 | 13.21 |
| Rotamer outliers | 0 | 12 | 6 | 8 |
| **Ramachandran plot** | | | | |
| Favoured (%) | 97.08 | 97.01 | 95.57 | 97.41 |
| Allowed (%) | 2.81 | 2.88 | 4.43 | 2.47 |
| Disallowed (%) | 0.11 | 0.11 | 0.00 | 0.12 |
| **EMDB Code** | EMD-11799 | EMD-11800 | EMD-10704 | EMD-12311 |
| **PDB Code** | 7AIN | 7AIO | 6Y5V | 7NGB |

**Table 1** (continued)

|  | KCC3b-WT (MSP1E3D1, NaCl) | KCC1Δ19 (Digitonin, NaCl) |  |  |
|---|---|---|---|---|
| Construct | KCC3b wild type (M1-S1099) | KCC1a (L20-S1085) |  |  |
| **Data collection** |  |  |  |  |
| Microscope | Titan Krios (OPIC, UK) | Titan Krios (MRCEF, UK) |  |  |
| Detector | K2 Summit | K3 |  |  |
| Voltage (kV) | 300 | 300 |  |  |
| Magnification | 130,000 | 105,000 |  |  |
| Collection mode | Counting | Counting (superresolution) |  |  |
| Electron exposure (e/Å$^2$) | 52.5 | 40 |  |  |
| Number of frames | 40 | 45 |  |  |
| Pixel size (Å) | 1.07 | 0.425 |  |  |
| Defocus range (μm; steps) | −1.25 to −2.75 (0.25) | −0.8 to −2.3 (0.3) |  |  |
| Number of movies | 2,362 | 18,446 |  |  |
| Phase plate used | No | No |  |  |
| **Data processing** |  | Reference | Subclass 1 | Subclass 2 |
| Initial Number of particles | 1,169,668 | 6,592,589 | – | – |
| Number of particles into 3D classification | 766,598 | 1,435,729 | – | – |
| Symmetry | C1 | C2 | C1 | C1 |
| Number of particles used for 3D refinement | 766,598 | 818,813 | 76,768 | 76,330 |
| Map resolution (Å; FSC threshold = 0.143) | 4.5 | 3.1 | 3.7 | 3.7 |
| Resolution range (Å) | 3.8–42.6 | 2.7–9.6 | 3.4–10.9 | 3.2–41.3 |
| Map sharpening B-factor (Å$^2$) | −148.7 | −165 | −127 | −121 |
| **Refinement** |  |  |  |  |
| Model resolution (Å; FSC threshold = 0.5) | – | 3.3 | 4.2 | 4.0 |
| **Model composition** |  |  |  |  |
| Non-hydrogen atoms | – | 13535 | 13514 | 13504 |
| Protein residues | – | 1734 | 1745 | 1751 |
| Ligands | – | 14 | 10 | 11 |
| **R.M.S.D** | – |  |  |  |
| Bond lengths (Å) | – | 0.006 | 0.008 | 0.008 |
| Bond angles (°) | – | 0.785 | 0.960 | 0.912 |
| **Validation** | – |  |  |  |
| MolProbity score | – | 1.63 | 1.78 | 1.70 |
| Clash score | – | 7.23 | 9.05 | 8.22 |
| Rotamer outliers | – | 10 | 19 | 13 |
| **Ramachandran plot** | – |  |  |  |
| Favoured (%) | – | 96.48 | 96.42 | 96.65 |
| Allowed (%) | – | 3.52 | 3.58 | 3.35 |
| Disallowed (%) | – | 0.00 | 0.00 | 0.00 |
| **EMDB Code** | EMD-11805 | EMD-11801 | EMD-11802 | EMD11803 |
| **PDB Code** | N/A | 7AIP | 7AIQ | 7AIR |

### N-terminal domain arrangement in phospho-mimetic KCC3b-PM

A unique feature observed in the phospho-mimetic KCC3b-PM structure is the presence of pronounced density for a short stretch of the otherwise unstructured N-terminus, a domain that has not been resolved in any of the previously published CCC structures. Distinct helical-shaped electron densities are visible near the intracellular cavity created by the hinged helices TM1a and TM6b in both chains of the KCC3 dimer (red surfaces in Figs 1B and 2A), which allowed us to build approximately 30 amino acids of the N-terminus,

assisted by the Rosetta FastRelax protocol (Conway *et al,* 2014) (Appendix Fig S5). This domain of the transporter folds into a triangular-shaped arrangement with two short helices and an unstructured portion forming the sides of the triangle (Fig 2F). The tertiary structure of this region is stabilized by multiple interactions with intracellular loops and several TM helices.

Electrostatic surface representations of KCC3b in absence (Fig 2B) or presence of the N-terminus (Fig 2C) illustrate that this segment is wedged between CTD and TMD and partially inserted into the inward-open vestibule created by the helical arrangement of TM2, TM7 and the helices TM1a and TM6b of the transporter core domain (Fig 2A–C). The direct contact with the ion coordinating helices TM1a and TM6b results in a visible obstruction of the ion access path from the intracellular side (Fig 2C inset), suggesting that this domain arrangement could play a role in controlling transport activity. While this structure was obtained under $K^+$-free conditions (Fig EV2A), we see the same configuration of the N-terminus in a lower resolution structure of KCC3b-PM in presence of $K^+$

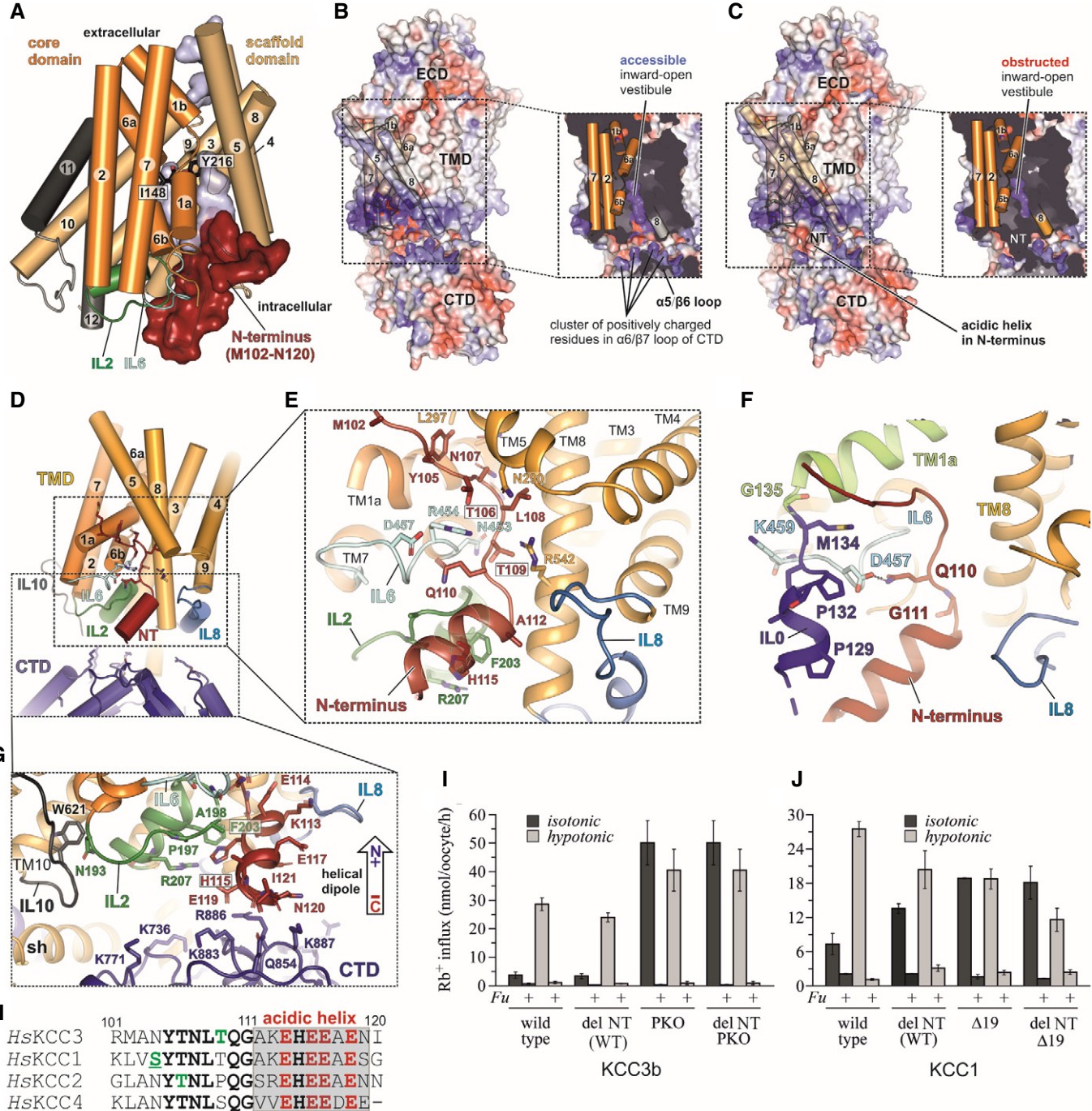

**Figure 2.**

**Figure 2.  Structural organization of the conserved N-terminal segment in KCC3b-PM and Rb⁺ uptake activity of NT-deleted variants in KCC1 and KCC3b.**

A   Cartoon representation of the KCC3b-PM TMD with N-terminal residues M102-N120 shown as red surface. The core domain (helices 1, 2, 6, 7) is represented in orange, the scaffold domain (helices 3–5, 8–10) in wheat. Residues I148 and Y216 involved in ion coordination are shown as black sticks.

B   Electrostatic surface representations of KCC3b-PM and slab view of the TMD (inset) in absence of N-terminal residues M102-M134. The surface is coloured by electrostatic potential (red, $-5$ kT e$^{-1}$; blue,$+5$ kT e$^{-1}$).

C   Same as in (B), but with N-terminal residues included, highlighting the obstruction of the inward-open vestibule.

D   Cartoon representation of TMD (yellow), intracellular loops (blue, green and light cyan), CTD (deep blue) and N-terminus (dark red).

E   Conserved polar residues (stick representation) in the N-terminus interact with IL6 (light cyan), IL2 (green), TM5 and TM8. IL0 is omitted for clarity.

F   N-terminal "triangular" extension formed by residues M102-M134 prior to TM1a (lime green helix). IL0 (blue helix) and NT (dark red) are stabilized by interactions to IL6 (light cyan).

G   Stabilization of acidic helix in N-terminus (red cartoon) by Pi-stacking to F203 in IL2 (green sticks) and backbone carbonyl interactions to positively charged residues in the CTD (deep blue sticks).

H   Sequence alignment of N-terminal segments from human KCCs with phosphorylation sites highlighted in green. Underscored letters represent phosphorylation sites confirmed experimentally in this study.

I   Rb⁺ uptake activity under isotonic and hypotonic conditions for KCC3b variants with internally deleted R101-I121, introduced to KCC3b-WT and KCC3b-PKO, respectively. Values are mean ($\pm$ SE) background-subtracted transport rates of 10 oocytes from 3-6 experiments in the presence or absence of 1.5 mM furosemide (indicated by +/− Fu).

J   Rb⁺ uptake activity under isotonic and hypotonic conditions for KCC1 (WT and Δ19), and the respective constructs with an internal deletion of residues K85-G105. Values are mean ($\pm$ SE) background-subtracted transport rates of 10 oocytes from 3 to 6 experiments in the presence (indicated by +*Fu*) or absence of 1.5 mM furosemide.

(Fig EV2D, Table 1). By contrast, the maps of KCC3b-WT, determined at 3.64 and 4.5 Å in detergent and MSP nanodiscs (Table 1 and Fig EV2B and C), do not show any electron density for the N-terminal portion. In line with an obstructed ion path, the KCC3b-PM map does not show any densities for cations coordinated by highly conserved residues of the cation-binding site: Y232, I148, P443 and T446 (Fig EV2E). This interpretation is further supported by density for a putative cation coordinated by the corresponding residues in the KCC1Δ19 structure without N-terminally "plugged" ion access path (Fig EV2F).

In agreement with the secondary structure predictions, the second half of the resolved N-terminal domain of KCC3b-PM folds into a short, "acidic NT helix" (Fig 2D–G). Four glutamates in this helix are highly conserved across the whole CCC family (Fig 2H and Appendix Fig S2) and could therefore play a role in electrostatically attracting this region to positive charges around the inward-facing vestibule and in adjacent C-terminal loops (blue coloured surface regions in Fig 2B). H115 in the centre of this short helix is conserved in all human KCCs, but replaced by glycine in N(K)CCs. It forms aromatic interactions with F203 and a cation-π with the side chain of a conserved arginine (R207) in intracellular loop 2 (IL2). The orientation of the N-terminal helix is favoured by the dipole moment from the combined effect of carbonyl oxygen atoms pointing towards the C-terminal end of the helix near the domain-swapped CTD from the other chain in the dimer (Figs 2G and EV2G). The negative end of the dipole is attracted by a cluster of conserved, positively charged side chains in the β7/α6 loop of the CTD (Fig EV2H). Notably, this particular N-terminal arrangement is most likely a unique property of the K⁺ transporting SLC12 family branch, assuming that all KCCs have a counter-clockwise twisted CTD. Due to the clockwise twist in *Dr*NKCC1, IL2 is brought into close proximity to an entirely different region of the CTD, which includes the VI/LTI/FYS motif at the end of the protein (Fig EV2I and J).

### Conserved N-terminal S/NY<u>T</u>NL<u>T</u>Q motif with phospho-regulatory sites

Residues M101-G111, located N-terminal from the acidic helix in KCC3b-PM, lack secondary structure and contain a S/NYTNLTQ

sequence motif, which is highly conserved only within the K⁺ branch of the SLC12 family (Fig 2H and Appendix Fig S2). Interestingly, this part of the N-terminus is in direct contact with the TMD region of the transporter and partially inserted into the intracellular vestibule formed by transmembrane helices TM1a and TM6b of the core and scaffold helices TM5 and TM8 (Fig 2D). Exposed backbone carbonyls and polar side chains in this region facilitate extensive interactions with charged side chains in these helices and with intracellular loops IL2 and IL6 (Fig 2E). The side chain of Q110 in this motif is sandwiched between these two intracellular loops, interacting with the backbone carbonyl of A198 (IL2) and the guanidinium group of R454 (IL6), which itself forms another H-bond with the backbone carbonyl of T109. The T109 side chain forms a hydrogen bond to the amino group of N453. The hydroxyl group of a second threonine within the motif (T106) is stabilized by H-bonds to the polar side chains of N290 (TM5) and R454 (IL6), while the backbone carbonyl of L108 interacts with the backbone carbonyl of N453. Interactions to TM1a are mediated via the side chain of N107, which forms a hydrogen bond with Q146 (TM1a).

Importantly, this conserved, unstructured stretch within the KCC3b-PM N-terminus is a hot spot for post-translational phospho-modifications (at residues highlighted in green in Fig 2H). N104 within this motif is conserved in human KCC2-4, but replaced in KCC1 by S88, which is phosphorylated in our mammalian expression system (Appendix Fig S6A). *In vitro* phosphorylation experiments (de Los Heros *et al*, 2014) suggest SPAK-mediated phosphorylation of T160 in KCC3a (corresponding to T109 in KCC3b) and T92 in KCC2a (corresponding to T106 in KCC3b). However, the purified KCC3b-PM protein used for our structural studies is not phosphorylated at these sites according to proteomics analysis (Appendix Fig S7A). Phosphate groups linked to these threonines would likely disrupt the tight network of interactions with IL6. These residues could thus act as a controllable switch for these putative inhibitory intramolecular interactions of the N-terminus, similar to the R-domain of CFTR, which is released from its inhibitory interaction with TM extensions upon phosphorylation (Liu *et al*, 2017).

To gain further insights into the functional significance of the N-terminal interactions observed in the KCC3b-PM structure, we

deleted a 20 amino acid stretch (shown in Fig 2H) in various KCC constructs and determined how $Rb^+$ uptake activities in *Xenopus* oocytes are affected. Surprisingly, we did not observe any significant differences to the respective parent constructs KCC3b-WT, KCC3b-PKO (Fig 2I and J) and KCC3b-PM (Appendix Fig S8). Because we see density for the N-terminal segment only in the KCC3b-PM mutant and not in the wild-type KCC3b structure, we suspected that this configuration could be stabilized by a conformation of the CTD that brings the β5/α5 and β7/α6 loops into close proximity of the N-terminus (Fig EV2G). This CTD conformation could be favoured by the phospho-mimetic mutations at the canonical sites, and we tested this hypothesis by introducing four charge-neutralizing point mutations (E114A/E116A/E117A/E119A) into the "acidic helix" of the N-terminus. We also investigated a second variant with multiple mutations in residues in the α5/β5 and α6/β7 loops of the CTD (Q854A/K883A/R886A/K887A) to disrupt the NT/CTD interface. Since we did not see any significant potentiation of $Rb^+$ uptake for the phospho-mimetic KCC3b-PM construct when introducing these second site mutations (Appendix Fig S8), we conclude that disruption of the putative contact does not rescue the phospho-inhibited phenotype of KCC3b-PM. However, we cannot exclude that expression or surface targeting is affected by these multi-site mutations; thus, further work with individual point mutations in the CTD/NT interface (eg. alanine scanning mutagenesis) is needed to clarify if a disruption of the interdomain contact might enhance transport activity of KCCs.

### Mobility differences of KCC3-PM and KCC3-PKO revealed by HDX-MS experiments

The striking N-terminal domain arrangement in the KCC3b-PM structure prompted us to investigate this mutant further and compare it directly to the constitutively active KCC3-PKO construct in HDX-MS experiments (Fig 3A and Appendix Fig S9A and B). Interestingly, the largest differences between phospho-inhibited KCC3b-PM and constitutive active KCC3-PKO are located in the region of residues 682-691, which correspond to the scissor helix that we found acting as a hinge region for the CTD movement of KCC1Δ19. Increased deuterium uptake for KCC3b-PKO suggests that the scissor helix is more dynamic and could indicate that this mutant also exhibits enhanced rotational flexibility of the CTD dimer. We hypothesized that the more rigid CTD arrangement in KCC3-PM could be linked to the presence of the N-terminal segment

stabilizing the CTD in the more twisted configuration, but the HDX-MS data for peptides from the N-terminal regions of KCC3b-PM and KCC3b-PKO do not suggest major differences in accessibility for deuterium exchange (Appendix Fig S9A and B). Additional regions with pronounced difference in H/D exchange rates are clustered in the outer lobe of the KCC3b CTD. The largest differences are within CTD helices β7, α8 and α10 and indicate more dynamic conformations in the KCC3-PM mutant, which, as mentioned, also exhibits larger deuterium uptake than the isotonically active PKO variant.

### 3D variability analysis for KCC1Δ19 and KCC3b-PM

Both 3D classification in the main EM data processing and HDX-MS suggested rigid body flexibility of KCCs with the scissor helices acting as pivots. Therefore, we used 3D variability analysis (3DVA) in Cryosparc package to assess the distributions of particles in each of KCC1Δ19 and KCC3b-PM datasets. As shown in Fig 3B and C for KCC1, two main modes of rigid body movement exist for both proteins (see also Movies EV1 and EV2). Due to the strong dimer interface interactions within the CTD inner lobes, the CTD dimer moves as a rigid body. On the level of TMD/ECD, each chain moves as a rigid body, thus breaking the overall two-fold symmetry (Movie EV1). These movements are centred around the scissor helices as pivot points, confirming our observations from HDX-MS that showed increased levels of deuterium exchange. The same rigid body groups exist in both KCC1 and KCC3, but the degree of movement of the CTD dimer is larger for KCC1, whereas the extent of ECD monomer separation as consequence of TMD tilting between the two chains is more pronounced in KCC3 (Fig 3B and C and Movies EV3 and EV4).

Another remarkable observation from 3D variability analyses is the coexistence of multiple conformations for the relative positioning of helices α8 in KCC1 (Figs 3D and EV3F) and α7 in KCC3b-PM (Fig 3E). The change in α7 positioning involves a 21° outward movement of the N-terminal end of α7, accompanied by a conformational change in the β6/α7 loop. A small conformational change in α7 and β6/α7 loop orientation is also seen for KCC1 (Fig 3D), but the reorganization of α8 in KCC1 is more dramatic. Helix α8 moves from a position close to the central β-sheet to a position further away from the TM region, over a distance of 30.6 Å (Cα–Cα distance of T926 in state 1 and state 2), by undergoing a 47° rotation around the β7 strand. The β7 strand is one of the regions exhibiting substantial differences in the HDX-MS spectra of KCC3b (Fig 3A), thus supporting the possibility of such conformational changes.

---

**Figure 3.  HDX-MS and 3D variability analysis highlight dynamic regions in KCCs.**

A  HDX-MS difference plot illustrating areas with altered deuterium exchange rates for PKO and PM of KCC3b. Relative fractional uptake differences (PKO minus PM Mutant) at three indicated time points are plotted against amino acid range covered by peptides. Negative uptake difference indicates an increased hydrogen exchange of PM. Positive uptake difference indicates a decreased hydrogen exchange of PM. All data points are mean from triplicates. Grey highlighted region indicates peptides with increased hydrogen exchange of the PM variant for an N-terminal portion not resolved in the cryo-EM structure.

B, C  Conformational dynamics of full-length KCC1Δ19 and KCC3b-PM structures inferred from normal modes (3D variability analysis. Movements of flexible regions are indicated by arrows. Extent of movement is quantified in Å or ° for KCC1Δ19 (values in blue or red) and KCC3b-PM (values in dark blue or dark red).

D, E  Co-existing alternative conformations of the CTD outer lobe helices α8 in KCC1Δ19 (D, violet and purple helices) and α7 in KCC3b-PM (E, blue and orange helices). Cα atom locations of T629 within the KCC1Δ19 α8 helices are highlighted in yellow. Alternative conformations of α10b of KCC1Δ19 are highlighted in teal and light teal cartoon representation.

F  Cartoon representation of KCC3b-WT CTD (pdb: 6MIY) with alternative α8 and α10b conformations, highlighted in violet and teal, respectively. Phosphorylated residue T940-P is shown as yellow sticks.

G  Cartoon representation of the CTD of *Ce*KCC1 (pdb: 6VWA) with alternative α8 and α10b conformations, highlighted in purple and light teal, respectively.

H  Cartoon representation of the CTD of *Dr*NKCC1 (pdb: 6NPJ).

Data information: Inner lobes of the CTDs in panels (D-H) are shown in light grey cartoon representation, outer lobes are highlighted in colours.

---

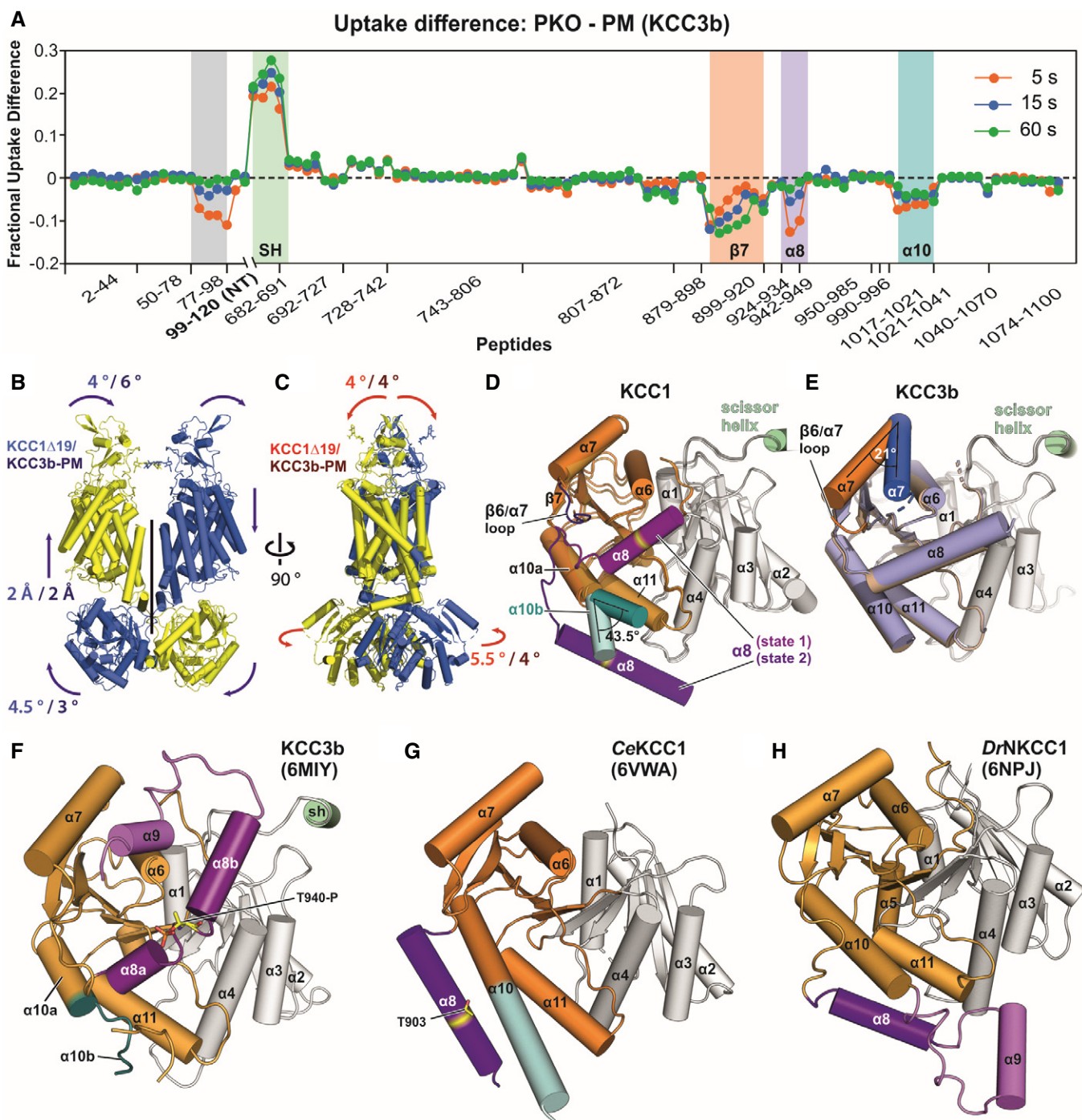

**Figure 3.**

In these two alternative conformations, α8 is either located above or below the α10 helix. Notably, also the latter helix α10 itself twists by 43.5° towards the α8 helix. The first state of the α8 helix close to the centre of CTD resembles our KCC3b-PM structure (Fig 3E) and the structure of wild-type KCC3b (Fig 3F) published by another group while this work was under review (Chi *et al*, 2020). A third group reporting the structure of KCC3a omitted this area for model building due to the limited local resolution (Xie *et al*, 2020). The

second state below α10 helix is more similar to the conformation of α8 in the isolated CTD from *Ce*KCC1 (Fig 3G) and to the conformation of the corresponding helix in *Dr*NKCC1 (Figs 3H and EV3G). This second state of α8 is stabilized by multiple polar interactions to a series of conserved glutamates in α11 (Fig EV3F), which creates a negatively charged patch on the surface of this region of the CTD (Appendix Fig S10B). The N-terminal end of α10 also exists in two alternative conformations according to the 3DVA results for KCC1

(α10b in Fig 3D). Again, this finding is in agreement with HDX-MS results (Fig 3A), and the fact that the structures from other groups exhibit substantially different conformations for α10, ie. α10b is partially unwound in the recent KCC3b structure (Fig 3F). Furthermore, the helical α10b structure reported for *Ce*KCC1 (Fig 3G) is sensitive to proteolytic digest (Zimanyi *et al*, 2020), thus supporting the possibility of a second, partially unwound structure. The conformational flexibility of the CTD outer lobe helices α7–α10 is well supported by our HDX-MS data, which also show increased deuterium uptake for these regions. Together, these results point at multiple dynamic regions within the CTD that undergo substantial conformational reorganization upon phosphorylation at the canonical threonine sites.

### Phosphorylation sites and nucleotide-binding motif in the CTD

Of the two canonical threonine sites targeted by SPAK/OSR-1 kinases, our cryo-EM maps only revealed the positions of site I (T940D in KCC3b-PM and T926 in KCC1) within the α8 helix of the CTD (Figs 4C and 5C). Due to the observed flexibility in the outer lobes of the CTD, a gap of ~80 amino acids in the models of both KCC structures prevented us from locating the position of site II (T983 in KCC1Δ19 and T991 in KCC3b-PM). This gap was also observed in the 2.2 Å X-ray structure of the isolated KCC1 CTD from *C. elegans* (Zimanyi *et al*, 2020).

Additional phosphorylation sites for KCC3b identified by proteomics analysis are shown in Fig 4 and Appendix Fig S7. S685-P is located in the scissor helix at interaction distance to E674 in TM12 (Fig 4A). Another site, T727-P (Fig 4B and Appendix Fig S7C) is present in α1 near the nucleotide-binding site (discussed below). S930-P is located close to the α8 hinge (Fig 4C and Appendix Fig S7E). Phosphorylation at this conserved residue could play a role for the alternative conformations we observed for the α8 helix in KCC1. Y1070-P located within α10, in proximity to a conserved cluster of prolines in the α10/β9 loop (Fig 4C), might be important for surface trafficking of the transporter, because phosphorylation by PKC at the corresponding site in KCC2 causes lysosomal degradation (Lee *et al*, 2007). A common feature of these phospho-sites in KCC3b is the presence of nearby positively charged arginine side chains, which likely play a role in counterbalancing the negative charge of the introduced phosphate groups. For some phosphorylation sites (e.g. S685 and Y1070) identified by MS-MS, we did not see clear densities for the phosphate and the introduction of it would cause clashes with residues in the vicinity. We have not built these in the model, and it is likely that the phosphorylation is only present in a subset of the particles.

We also mapped two sites uniquely present in KCC1: S734-P and S1050-P (Appendix Fig S6B and D). Interestingly, these sites are located within the inner lobe of KCC1 (Fig 5C) and replaced by non-conserved amino acids N748 and P1064 in KCC3b-PM (cyan spheres in Fig 6A). These unique phosphorylation sites in KCC1 are located near a conserved structural motif closely resembling the ATP-binding motif in bacterial universal stress proteins (USPs). DALI (Holm & Rosenström, 2010) searches identified the regulatory subunit TeaD (pdb: 3HGM) of the ectoine-specific transporter TeaABC from *Halomonas elongata* as closest hit (Fig EV4C), among a few other proteins with nucleotide co-factors. The osmo-regulatory transporter TeaD belongs to the family of tripartite ATP-independent periplasmic transporters (TRAP-T), evolutionarily placed between the families of secondary active and ABC transporters. Although the precise role of the regulatory subunit of this transporter is not fully understood, genetic deletion of the subunit potentiates transport and ATP binding promoted TeaD oligomerization *in vitro* (Schweikhard *et al*, 2010). Despite the high structural similarity between the inner lobe of the CTD to TeaD, there is very little sequence identity in the primary protein structures of KCCs and TeaD, which prevents traditional sequence alignments without structural restraints.

### ATP/ADP binding to inner lobe pocket of KCC1Δ19

Inspection of the electron density in this region of the KCC1 map suggests that either ADP or ATP are bound to the Rossmann-like fold formed by β-strands β1, β2 and β4 and helices α1, α3, α4 of the KCC1 inner lobe (Fig 5). Combined dipole charges from the converging N-terminal ends of the three helices and multiple side chains create a positively charged nucleotide-binding pocket (Fig 5B and Appendix Fig S10A). Weaker density for the γ-phosphate indicates that a mixture of ADP- and ATP-bound particles may contribute the observed densities (Fig EV4A). LC-MS analysis of denatured purified KCC1 protein confirmed the presence of ADP and ATP in the sample (Appendix Fig S11), nucleotides that are likely carried along from the cell lysate during purification, or introduced by the ATP/MgCl$_2$ wash step (see Methods for details). Similar to the ATP-binding mode in *He*TeaD (Fig EV4C, Appendix Fig S10E), the purine base is stabilized by polar interactions with the backbone peptide to V730 in the β2-strand of the CTD (Fig 5D). While this residue is universally conserved in all human KCCs and NKCCs (Fig EV4B and D, Appendix Fig S2), two lysines (K699 in the β1-strand and K707 in the β1/α1 loop of KCC1) that stabilize the Pγ of ATP are present only in KCC1 and KCC3, respectively. Sequence alignment of these regions involved in phosphate coordination shows that KCC4 only has the second lysine (K707 in KCC4a), whereas two arginines in KCC2 (R679 in β1 and R777 in the β4/α4 loop of KCC2b) could engage into similar electrostatic interactions (Fig EV4B).

By contrast, no densities for ATP or ADP are visible in the map of KCC3b-PM, which could be the result of small sequence differences in KCC3b. As illustrated by the superposition of the inner CTD lobes of KCC1 and KCC3b (Fig 6A), the larger hydrophobic side chains in α3 of KCC3b (L784 and I788 in cyan) could interfere with the binding of the purine base. In particular, the presence of L784 in KCC3b creates a visibly smaller, positively charged cavity compared to V770 in KCC1 (Fig 6D and F vs. Fig 6C and E and Appendix Fig S10C). Furthermore, two tyrosine residues (Y708 and Y797) involved in nucleotide binding by KCC1 correspond to H722 and N811 in KCC3b-PM. Another possibility is that nucleotide binding is sensitive to the phosphorylation state of T727 in α1 of KCC3b (Fig 6A). This site is located near K713 and K721, and small changes introduced by its phosphorylation could disrupt favourable interactions with the γ-phosphate of ATP. Notably, we observed phosphorylation for T727 in KCC3b by proteomics analysis (Appendix Fig S7D), but not for the corresponding T713 in KCC1 (Appendix Fig S6G).

### MD simulations with ATP indicate tight binding to KCC1 and weaker binding to KCC3b

To investigate whether the small differences in the binding pocket of KCC1Δ19 and KCC3b-PM can affect the stability of ligand

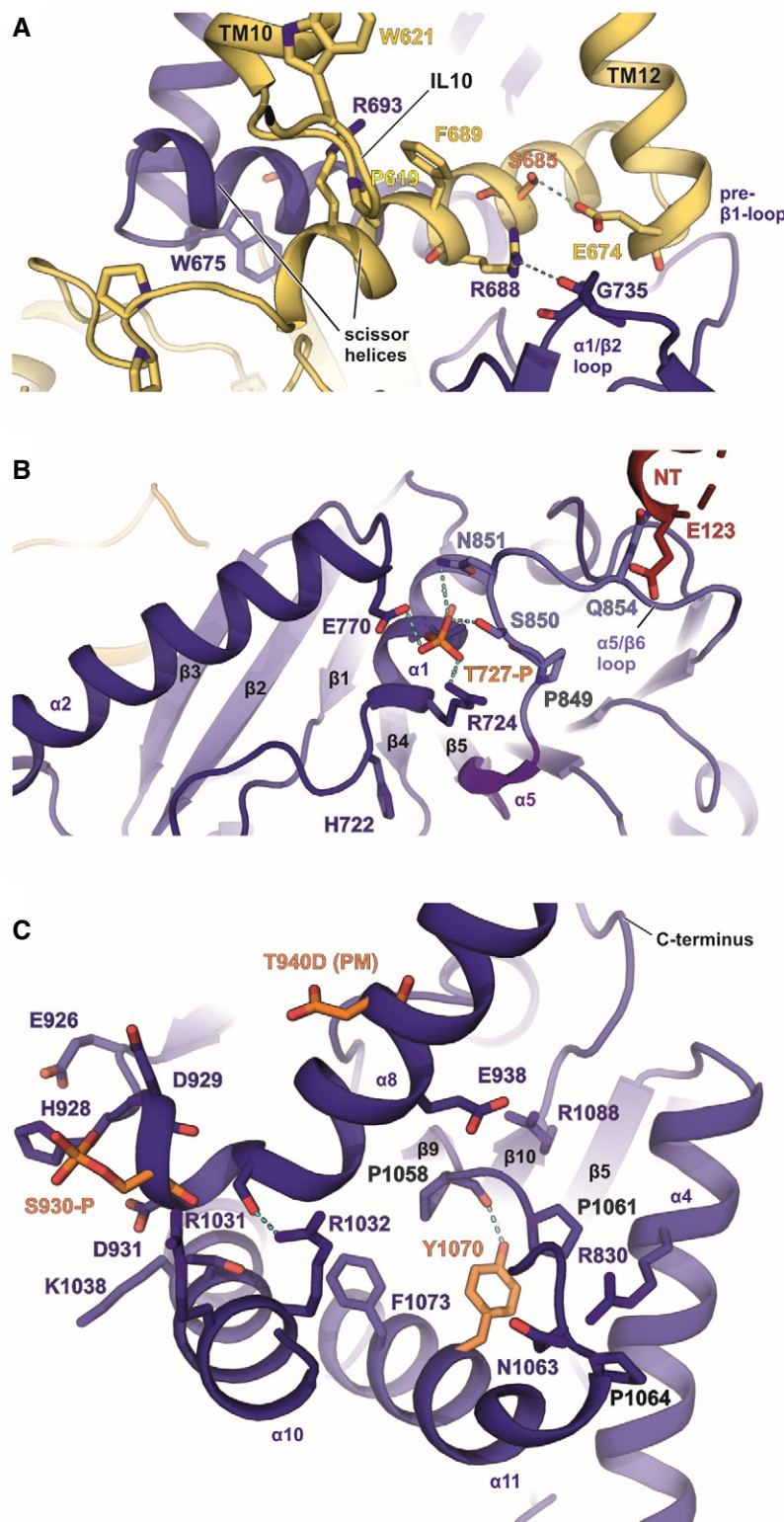

**Figure 4.  Location of phosphorylation sites in the C-terminal domain of KCC3b-PM.**

A   Location of S685-P in the scissor helix of the CTD.
B   T727-P in α1 near the ATP/ADP-binding site.
C   Location of T940D and S930-P in α8 and Y1070 in α10. Phosphorylated residues according to MS-MS data are shown in orange. Arginines and prolines in the vicinity are highlighted with blue and grey labels, respectively.

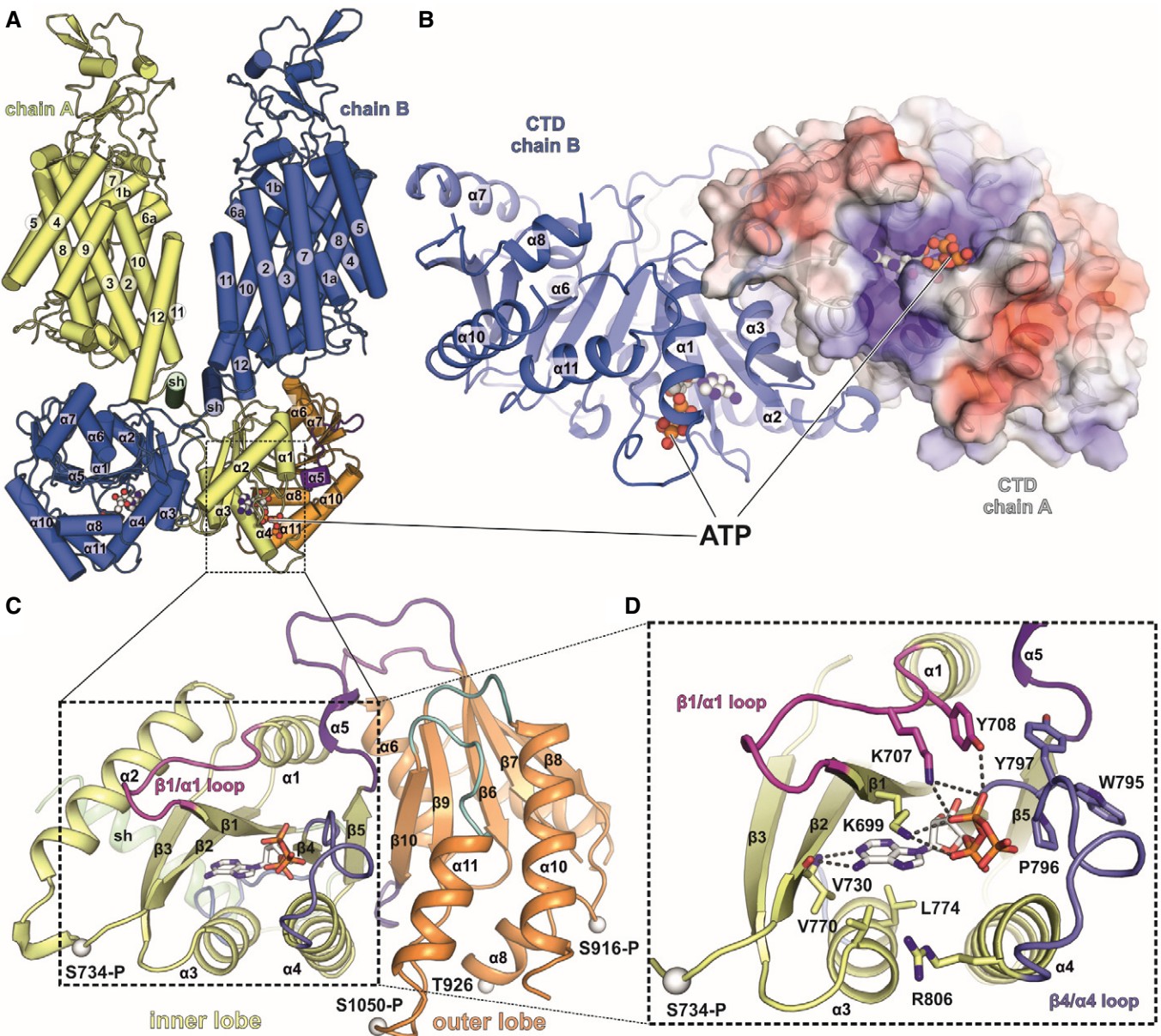

**Figure 5. Location of ATP/ADP-binding motif and phosphorylation sites in the C-terminal domain of KCC1Δ19.**

A Front view of full-length KCC1Δ19 with ATP (spheres) bound to both CTDs in the dimer. Overview of the KCC1 structure in cartoon representation.

B Intracellular view onto the CTD dimer of KCC1Δ19 with ATP bound to each chain. Chain A is shown as electrostatic surface coloured by electrostatic potential (red, $-5$ kT e$^{-1}$; blue,$+5$ kT e$^{-1}$), chain B as cartoon.

C C-terminal domain organization of KCC1, with scissor helix (green), inner lobe (yellow) and outer lobe (orange). Cαs of phosphorylated residues are shown as white spheres.

D Detail of the inner lobe with ATP bound via backbone interactions to V730 and side chain interactions with two lysines: K699 and K707 in the β1/α1 loop (shown in pink) and additional residues in the β4/α4 loop (shown in slate blue).

binding, we carried out multiple MD simulations of KCC1 and KCC3b (Fig EV5). ATP stayed in position with very minor ligand displacement for MD runs with the KCC1 dimer (TMD+CTD, Fig EV5A), and the same behaviour was also observed for runs with the isolated, monomeric CTD of KCC1 as a simplified MD system (Fig EV5B). The bidentate interactions of the purine base to V730

and the interactions of Y708 with the β- and γ-phosphates remained stable (present for greater than 90% of the simulation) for both chains in the KCC1 dimer over the duration of the 300 ns run (Appendix Fig S12A and B). These interactions also remained stable in simulations of the KCC1 CTD alone for > 90% of the run (Fig EV5D).

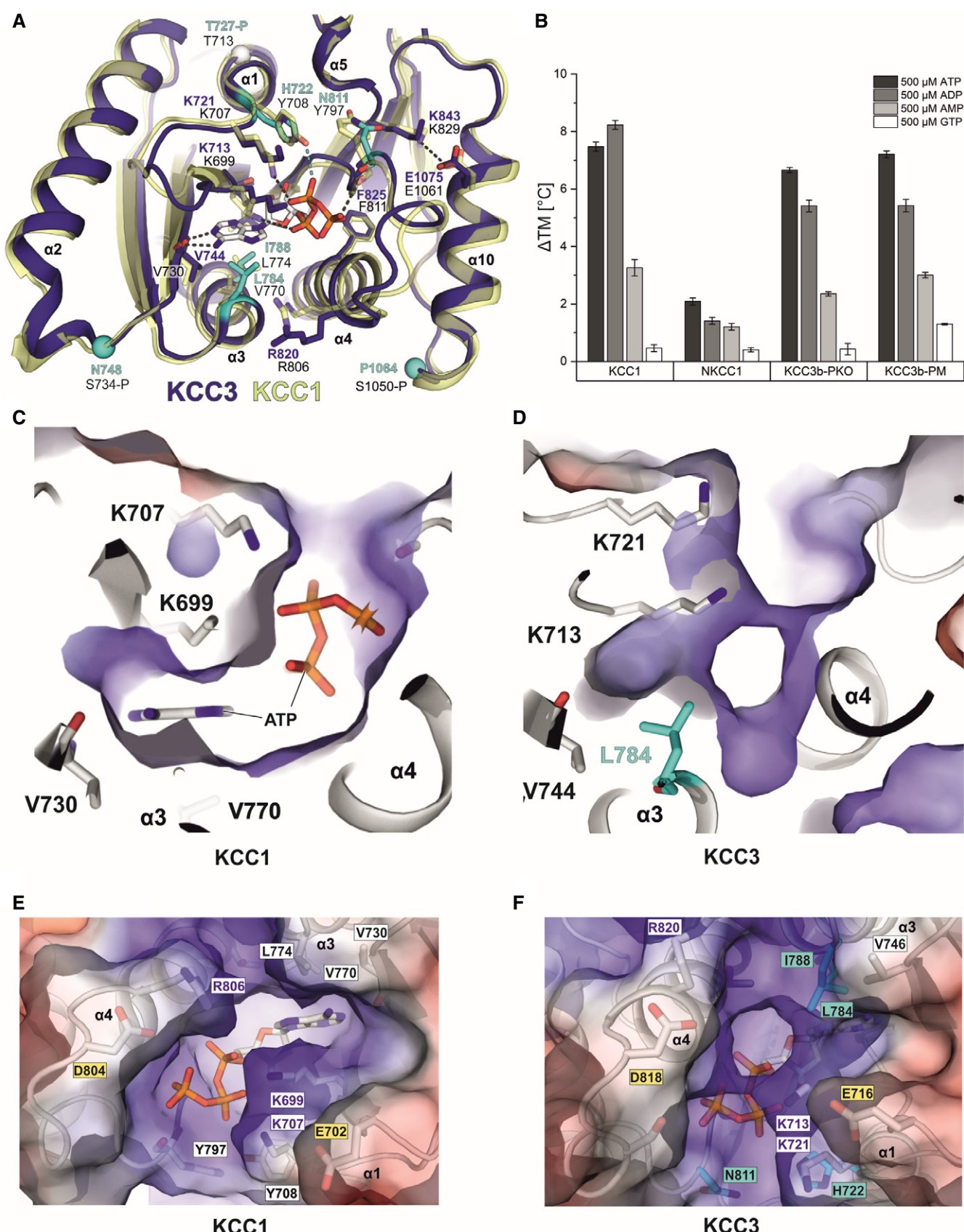

**Figure 6.**

**Figure 6.  Comparison of nucleotide interactions of KCC1 and KCC3b by molecular dynamics simulations.**

A      Superposition of the inner lobe of KCC1 (construct Δ19 in yellow cartoon representation) and KCC3b (construct PM in deep blue cartoon representation) with sequence differences in KCC3b highlighted in cyan.
B      Bar chart illustrating the extent of thermostabilization (ΔTM shift in melting temperature determined by nanoDSF) by different nucleotides at 500 μM of human KCC1, NKCC1, KCC3b-PKO and KCC3b-PM. Values are mean (± SE) from triplicates for each condition.
C–F  Electrostatic surface representation of the ATP-binding pocket in KCC1 (C, E) and the respective region in KCC3 (D, F). The surface is coloured by electrostatic potential (red, $-5$ kT e$^{-1}$; blue,$+5$ kT e$^{-1}$). Labels in cyan highlight residues different in KCC3, yellow labels indicate residues with potential roles in Mg$^{2+}$ coordination or ATP hydrolysis. White labels highlight residues with a major role in nucleotide binding.

On the other hand, simulations with ATP introduced into the respective KCC3b-PM CTD showed a substantially less stable interaction (Fig EV5C and E). Specifically, in run 1 of KCC3 the interaction with the two lysine side chains was only preserved in the beginning of the MD run and a destabilization after 150 ns leads to a disruption of the interactions with the β- and γ-phosphates, and ultimately to a loss of the ATP binding (Movie EV5, Movie EV6 and Appendix Fig S13C). The bidentate interaction to V744 in KCC3 is only continuously maintained, > 90% of the time of the simulation, for one single run (Fig EV5E). One of the most significant differences in the sequence that may explain the lack of ATP in the KCC3 map is H722, whose side chain is only seen to be interacting with the nucleotide greater than one-third of the time for two of the three simulations conducted (Fig EV5E, Appendix Fig S13D). By contrast, the corresponding residue Y708 in KCC1 forms a direct interaction to the β-phosphate of ATP and a water-mediated interaction for more than three-quarters of the time in all our KCC1 simulations. Furthermore, the 2' an 3' hydroxyl groups of the ribose in ATP show a number of additional water-mediated interactions to residues in the β4/α4 loop (G794, W795) and L697 (β1) of KCC1 (Fig EV5D), which are entirely absent for KCC3 (Fig EV5E). Together, these MD results confirm overall weaker ligand-protein interactions for KCC3b and might therefore explain the absence of ATP from our KCC3 cryo-EM maps.

In one of the MD runs of KCC1 (CTD only, Fig EV5D), D701 in the β1/α1 loop interacts with a water molecule near the γ-phosphate of ATP. This aspartate residue is conserved across the KCC subfamily (Fig EV4B), and *He*TeaD has an aspartate (D10) in the slightly shorter β1/α1 loop as well (Fig EV4C). Furthermore, the β4/α4 loop on the other side of the pocket contains an aspartate (D804) positioned in proximity to the γ-phosphate, which is conserved in members from both branches of the family (D816 in *Dr*NKCC1, Fig EV4D). It could potentially play a role in Mg$^{2+}$ coordination, similar to the aspartate in the Walker B motif (φφφφDE where φ is a hydrophobic amino acid) in ABC transporters (ter Beek *et al*, 2014). Next to these aspartates, the loops also contain conserved glutamates (E702 in the β1/α1 loop and E803 in the β4/α4 loop of KCC1, Fig EV4A). In analogy to the model proposed for ABC transporters (Priess *et al*, 2018), the glutamate adjacent to the Mg$^{2+}$ coordinating aspartate could serve as a transient proton acceptor ("general base") to activate a water for a subsequent nucleophilic attack on the γ-phosphate. Therefore, these conserved structural elements merit further investigation in future studies addressing the question whether the CTDs of the SLC12 transporters can catalyse ATP hydrolysis and whether this drives a conformational transition.

### Thermal stabilization of full-length CCCs by adenine nucleotides

To study the interaction of KCC1 with a range of different nucleotides, we carried out nanoDSF experiments with purified KCC1Δ19

and observed a concentration-dependent thermal stabilization by ATP and ADP (Appendix Fig S14A and B). The protein was also stabilized to a lesser extent by AMP, while GTP did not increase the melting temperature of KCC1 at all (Fig 6B). A slightly higher stabilization of the protein by ADP (9°C vs 7°C by ATP) could indicate a higher affinity for the dinucleotide.

Intriguingly, a recent study utilizing a photo-clickable ATP mimetic for the identification of interactors in the membrane proteome (Jelcic *et al*, 2020) suggests binding of ATP to several members of the SLC12 family, including members from both branches (KCC1, KCC3, NKCC1, CIP). Furthermore, the presence of unmodelled densities in the deposited cryo-EM maps of *Dr*NKCC1 CTD suggests ATP or ADP binding to the structurally highly conserved fold in the inner CTD lobe (Fig EV4D, Appendix Fig S10D). These findings prompted us to study nucleotide binding to a few additional members, i.e. human NKCC1 and several KCC3b variants in thermal shift experiments (Fig 6B). In line with our structural findings, we observed a significant thermostabilization for purified NKCC1 in presence of ADP and ATP, but did not observe a significant shift in melting temperature for WT KCC3 (Appendix Fig S14F). Although the latter observation fits well with our MD simulations and with the absence of density for a nucleotide in our KCC3b-PM EM maps, it is surprising to observe a thermal shift for variants KCC3b-PM and KCC3b-PKO. Another interesting finding from these experiments is that the Na$^+$ dependent transporter NKCC1 shows a lower stabilization by all four investigated nucleotides. Together with the structural data, these results allow us to conclude that nucleotide binding is a property shared between several members of the SLC12 family, hinting at a fundamental role for transport activity.

## Discussion

Our new structures of potassium-coupled chloride transporters unveil two important structural features:

- An inhibitory segment in the N-terminus of KCC3b-PM.
- An ATP/ADP-binding site in the CTD of KCC1Δ19.

Both known and new phospho-regulatory sites identified in this work are also clustered around these regions, implying that we have uncovered the architecture of key areas relevant for modulation of transport activity. Obstruction of the ion access path by an N-terminal segment undergoing reversible phosphorylation is a regulatory mechanism that has not been reported for any other transporter within the solute carrier superfamily. However, a similar mechanism has been described for CFTR through the involvement of the acidic R-domain (Liu *et al*, 2017). Although our deletion variants did not show enhanced transport activity, it must be noted that our experiments do not precisely reflect the transport mode of KCCs

under native conditions. A block by the N-terminus from the cytoplasmic side is more likely to occur under conditions that run the transporter in "forward mode", and the deletion variants might still exhibit enhanced transport activity under such conditions. Due to these limitations, we believe that our results do not preclude an inhibitory role of this domain *in vivo*, and future studies with genetically modified mice could be helpful to investigate this possibility further in a native environment.

From a physiological perspective, it would make sense for KCC activity to be under very strict control mechanisms. Tight regulation of potassium efflux is particularly crucial for cells to minimize energy depletion that could result from enhanced sodium/potassium ATPase activity, which maintains high intracellular $K^+$ under consumption of cellular ATP. Control by this inhibitory "N-terminal plug" (similar to the ball-on-a-chain N-type inactivation mechanism of certain potassium channels; Antz & Fakler, 1998) could be particularly relevant in excitatory cells where intracellular $K^+$ and $Cl^-$ concentrations are major determinants of the activation threshold. Consequently, rapid control over ion efflux by KCCs is essential to avoid changes in membrane potential as is now well-known in the case of KCC2, a CCC that play a key role in GABAergic inhibition (Pisella *et al*, 2019; Watanabe *et al*, 2019). The high degree of conservation in the N-terminal "plug" sequence suggests that an inhibitory configuration similar to our KCC3b-PM mutant could also exist in KCC2 with potentially important implications for neurodevelopmental physiology and pathology. This configuration is supported by two recent studies that were published while the current work was under review (Chi *et al*, 2020; Xie *et al*, 2020). The related papers report cryo-EM structures of human KCC2/3/4 with densities for a similar inhibitory N-terminal peptide. Functional data from these groups suggest potentiation of activity for deletions (Chi *et al*, 2020) or point mutations (Xie *et al*, 2020) by disrupting the auto-inhibitory action of the N-termini.

Our second major finding, the ATP/ADP-binding site in the inner CTD lobe of KCC1, together with the thermal shift experiments strongly suggest that NKCC1 and KCC3 phosphovariants also bind ADP or ATP. This observation is also in line with the results of a recent membrane proteome screening study in which a photo-clickable ATP mimetic was used as nucleotide (Jelcic *et al*, 2020). Our MD simulations demonstrate that the binding of ATP is sensitive to even small changes in the local environment. These results indicate that the stability of nucleotide binding could also be modulated by long-range effects from conformational changes in the outer lobe of the CTD, perhaps in response to a change of phosphorylation state at the canonical regulatory sites. The structural similarity to the ATP-binding motif from a halophilic bacterium suggests that we might have discovered an evolutionary conserved, ancient property for osmo-regulatory transporters. Interestingly, the binding mode of ATP is similar to certain ATPases with R-finger motif in which the transition state is stabilized by positively charged side chains (K699 and K707 in KCC1), which neutralize the $P_\gamma$ and promote hydrolysis.

The difficulty of manipulating intracellular nucleotide concentrations in living oocytes prevented us from investigating the functional effects of ATP or ADP binding on KCC1/3 transport activity in $Rb^+$ uptake experiments. *In vitro* transport assays with KCCs reconstituted into proteoliposomes could be an elegant way to address this important question in the future, and the feasibility of such assays has been recently demonstrated for KCC1 (preprint: Zhao *et al*, 2020). Future work should also address the question of whether the bound nucleotide acts as a "built-in" co-factor for any of the KCC-modifying kinases and whether phosphorylation at certain sites will disappear in response to disrupting ATP binding by site-directed mutagenesis.

Interestingly, we also identified structural elements in the binding pocket that could potentially play a role in catalysing the hydrolysis of bound ATP. The findings could hint at the possibility that the transporter itself exhibits ATPase activity and thus provide energy to drive a large conformational change, like the separation of the CTD dimer. Although such movements reminiscent of ABC transporters are speculative for CCCs, some studies have already suggested that the CTD might reversibly split into monomers for KCC1 (preprint: Zhao *et al*, 2020), KCC2-4 (Xie *et al*, 2020) and NKCC1 (Monette & Forbush, 2012). On the other hand, results from studying NKCC1 in squid axons hint at nucleotide binding alone without hydrolysis of the bound ATP. Russell and colleagues reported that ATP and a range of adenine nucleotides including adenine (but not thymine or cytosine) were effective at increasing saturable $^3H$ bumetanide binding to NKCC1 in squid exosomes (Altamirano *et al*, 1988; Altamirano *et al*, 1990). Notably, the authors concluded that this observation must be independent from the effects of activating NKCC1 via phosphorylation, because the increased $^3H$ bumetanide binding occurred also in the absence of $Mg^{2+}$, which is essential for ATP hydrolysis to drive phosphorylation or conformational changes. These results by Russell et *al* rather hint at the possibility that binding of ATP or ADP might enhance activity via a different mechanism, potentially by increasing the overall structural integrity of the CTD, which might act as a stabilizing scaffold to anchor the substantial conformational changes concomitant with transport activity.

It is intriguing to speculate that cells might be taking advantage of the high abundance of ADP and ATP and utilize these nucleotides for the stabilization of the ubiquitously expressed cation chloride transporters. Such a potential role as "pharmacological" or "physiological chaperone" is supported by the substantial ATP- or ADP-independent increase that we observed in thermal stability as well as by our 3DVA results where the core formed by the inner lobes at the centre of the CTD dimer was found to be rigid and immobile compared to the rest of the transporters. Another possibility is that the conformational equilibrium between inward-facing, occluded and outward-facing conformation of CCCs is biased towards the inward-facing state and a high kinetic barrier must be crossed for a transition into the other states, making them relatively short-lived. In fact, most of the published CCC structures are captured in the inward-open state and the only structure solved so far in an outward-open state is KCC1 bound to inhibitor VU0463271 (preprint: Zhao *et al*, 2020), locking the transporter in this state. Similarly, bumetanide might bind to the extracellular domain of NKCC1 in the outward-open state, and the reported increase in $^3H$-binding sites in the presence of adenine nucleotides could hint at a higher apparent affinity for the antagonist, caused by a shift in the conformational equilibrium towards the binding-competent outward-facing state. This could be coupled to hydrolysis of ATP or be driven by the binding of the nucleotide itself, similar to conformational changes induced in certain pseudokinases lacking hydrolytic activity.

Understanding the functional relevance of the highly conserved ATP/ADP-binding motif merits further investigation and holds potential for exploitation in drug discovery. Likewise, the finding of N-terminal obstruction of the ion access path could enable the development of peptide drugs or small molecules mimicking the action of the KCC3 N-terminus. For example, selective KCC1 or KCC3 antagonists would be suitable for the therapy of sickle cell anaemia, to prevent overactive transporters from dehydrating erythrocytes and contributing to HbS polymerization (Brown *et al*, 2015). Vice versa, drugs interfering with the auto-inhibitory intramolecular interaction in KCC3 could be useful for enhancing transport activity of the closely related KCC2 transporter. Such direct potentiators of Cl⁻ transport activity, targeting the solute carrier itself (as opposed to targeting the regulating kinases), are considered promising drug candidates for the treatment of severe human diseases, ranging from neurodevelopmental disorders associated with epileptic seizures (Fragile X Syndrome, Rett Syndrome, Down Syndrome) to GABA-deficient conditions including schizophrenia, autism and chronic neuropathic pain.

## Conclusion

Our multidisciplinary study provided new insights into the structural determinants of phospo-regulation in both N- and C-termini of KCCs and allowed us to discover a previously unknown nucleotide-binding pocket in the large C-terminal domain. These findings shed light into the complex mechanisms of transport modulation and unlock new avenues towards drug development for this important class of ion transporters.

# Materials and Methods

### Molecular biology, virus production and protein expression

Full-length and Δ19-deleted human KCC1 (isoform A) and KCC3 isoform B were cloned from the mammalian gene collection into pHTBV C-terminally tagged twin-strep, 10-His vector with and without GFP. Full-length NKCC1, phospho-mimetic (S45D, T940D, T997D) and phospho-knockout (S45A, T940A, T997A) constructs of KCC3b were synthesized (GenScript, Twist Bioscience) and subcloned into pHTBV C-terminally tagged twin-strep, 10-His vector with GFP. Baculoviruses for these constructs were generated following the standard protocol outlined in (Mahajan *et al*, 2021). Baculoviral DNA from transformed DH10Bac was used to transfect Sf9 cells to produce baculovirus particles, which were then amplified with Sf9 cells grown in Sf-900™ II media supplemented with 2% foetal bovine serum (Thermo Fisher Scientific). Cells were incubated on an orbital shaker for 65 h at 27°C. Cultures were centrifuged at 900 *g* for 10 min to harvest the supernatants containing the viruses. 1 l of Expi293F™ GnTI⁻ cell cultures in Freestyle 293™ Expression Medium (Thermo Fisher Scientific) was infected with P3 baculovirus (3% v/v) in the presence of 5 mM sodium butyrate. Cells were grown in a humidity-controlled orbital shaker for 48 h at 37°C and 8% $CO_2$ before being harvested by centrifugation at 900 *g* for 10 min, washed with phosphate-buffered saline and pelleted again prior to flash freezing in liquid nitrogen (LN2), then stored at −80°C until needed.

### Protein purification

Whole cell pellets expressing the constructs of NKCC1, KCC1 and KCC3b were resuspended to a total volume of 50 ml per 15 g of cell pellet with buffer A (150 mM NaCl, 20 mM HEPES pH 7.5) supplemented with 0.7% w/v Lauryl Maltose Neopentyl Glycol, LMNG (Generon), and 0.07% cholesteryl hemisuccinate, CHS (Generon). The cells were solubilized at 4°C for 1 h with gentle rotation. Cell debris was pelleted at 50,000 *g* for 30 min. The clarified lysate was added to 0.5 ml bed volume of Strep-Tactin SuperFlow (IBA) per 50 ml of lysate and allowed to bind at 4°C for 1 h. The resin was collected on a gravity-flow column and washed with buffer B (buffer A with 0.003% w/v LMNG and 0.0003% w/v CHS), and then with buffer B with 1 mM ATP and 5 mM $MgCl_2$ added. Protein was eluted with 7 CV of buffer B containing 5 mM D-desthiobiotin followed by tag-cleavage by TEV protease overnight and reverse purification. For LMNG/CHS condition, the samples were subjected to size exclusion chromatography pre-equilibrated with Buffer B (buffer A with 0.04% digitonin for digitonin condition; buffer A for nanodisc condition). Peak fractions were pooled and concentrated to 5 µM for LMNG/CHS and nanodisc samples and 50 µM for digitonin samples for subsequent experiments.

Nanodisc sample was prepared similarly to the detergent samples with a few exceptions. After washing the protein-bound Strep-Tactin resin, buffer A supplemented with 0.5% LMNG, 0.05% CHS and 0.125% soy azolectin (Sigma) was added to a final LMNG concentration of 0.2% and purified MSP1E3D1 protein to a final concentration of 0.5 mg/ml. The slurry was incubated on a rotating wheel for 15 min; then, 100 mg of washed Biobead SM-2 per ml resin was added, followed by further incubation for a minimum overnight. Subsequent purifications were then performed with buffer A.

### Cryo-electron microscopy sample preparation, data collection and data processing

All samples were frozen on Quantifoil Au R1.2/1.3 300-mesh grids glow discharged for 30 s, with plunge freezing performed on Vitrobot Mark IV (Thermo Fisher Scientific) chamber set to 80–100% humidity and 4°C. For LMNG/CHS and MSP1E3D1 conditions, blotting time was set to 1.0–1.5 s, and for digitonin conditions, it was set to 3.5–5.0 s after 30 s wait time.

The facilities and data collection strategies for each dataset are detailed in Table 1, and the data processing workflow for digitonin datasets of KCC3b-PM and KCC1Δ19 is detailed in Appendix Figs S3 and S4, respectively. Full details on data collection and processing are stated in the Appendix Supplementary Methods. The potassium-free digitonin datasets of KCC3-PM and KCC1Δ19 as well as LMNG/CHS and MSP1E3D1 datasets of KCC3b-WT were primarily processed with Cryosparc 2.11.0 with some initial steps done on Relion 3.0.8. Potassium-saturated digitonin dataset of KCC3b-PM was primarily processed with Relion 3.0.8.

For the model generation of the C-terminal domain (CTD) of KCC3, a Rosetta-generated model based on *Dr*NKCC1's CTD was used. For the transmembrane domain (TMD) of KCC3, a monomer model of *Dr*NKCC1's TMD (PDB: 6nph) was used, and the sequence was substituted with KCC3's TMD using CHAINSAW (Stein, 2008). These models were combined and further built in Coot (Emsley & Cowtan, 2004). They were refined using Phenix real space refine

(Afonine *et al*, 2018), and geometry of the models was verified in Phenix (MolProbity) (Klaholz, 2019).

The model for KCC3 was used as reference for KCC1, and sequence substitution was performed with CHAINSAW. Manual building and refinement was performed in Coot. Models were refined using Phenix real space refine, and geometry of the models was verified in Phenix (MolProbity).

### Rb$^+$ flux assays

Synthetic genes for KCC1 and KCC3 variants with N-terminal HA-tag were cloned into pPOL vector for *in vitro* transcription (Marcoux *et al*, 2019). Defolliculated stage V–VI *Xenopus laevis* oocytes were microinjected with 50 ng cRNA and maintained in Barth medium for 3 days at 18°C in the presence of 1.5 mM furosemide. Water-injected oocytes were used as controls. Before the transport assay, furosemide was removed through several washes in plain Barth medium. Carrier activity was assessed at room temperature through Rb$^+$ influx assays under isotonic and hypotonic conditions, i.e. by incubating oocytes for 1 h in a hypotonic solution (125 mOsM) or in an isotonic solution (200 mOsM) and reincubating them afterwards for 45 min in an isotonic salt-added physiological solution (7 mM Rb$^+$, 86 mM Cl$^-$) in the presence or absence of 1.5 mM furosemide. At the end of flux assays, oocytes were washed several times in a refrigerated Rb$^+$-free solution, lysed in pure nitric acid and assayed for Rb$^+$ content (1 oocyte/sample) by atomic absorption spectrophotometry (Varian AA240). Transport data for oocytes are expressed in this work as mean ($\pm$ SE) background-subtracted transport rates in 10 oocytes among 3–6 experiments.

### Hydrogen–deuterium exchange mass spectrometry (HDX-MS)

KCC3b-PKO and KCC3b-PM were diluted to 3.5 mg/ml using equilibration buffer which contains 25 mM HEPES pH 7.4, 150 mM NaCl and 0.002% LMNG. 5 μl of each sample was incubated with 50 μl of D$_2$O buffer (25 mM HEPES pD 7.4, 150 mM NaCl, 0.002% LMNG) for a time course of 5, 15 and 60 s, then quenched by 55 μl of ice-cold quenching solution (25 mM HEPES pH 1.9, 150 mM NaCl, 0.002% LMNG). 80 μl of quenched samples was loaded into nanoACQUITY UPLC System (Waters corp.) and online digested by Enzymate™ BEH Pepsin Column (2.1 × 30 mm, Waters corp.) at 20°C. The digested peptides were trapped onto a BEH C18 trap column (1.7 μm, 2.1 × 5 mm, Waters corp.) and separated by BEH C18 analytical column (1.7 μm, 1 × 100 mm, Waters corp.) with a linear gradient of buffer B (acetonitrile with 0.1% formic acid) from 3 to 35% at a flow rate of 40 μl/min.

Mass spectra were acquired using Synapt G2-Si HDMS mass spectrometer (Waters Corp.) in positive mode. MS/MS spectra were acquired in MS$^E$ mode. Peptides from un-deuterated samples were identified by ProteinLynx Global Server 2.5.1 (Waters Corp.), and HDX data were analysed by DynamX 3.0 (Waters Corp.). Relative fractional uptake was calculated by dividing the measured deuterium uptake by the theoretically maximum deuterium uptake.

### Phosphorylation site identification by proteomics analysis

To identify phosphorylation sites on KCC1 and KCC3b, samples were separated by SDS–PAGE. Targeted bands were cut, reduced with 25 mM TCEP, alkylated with 55 mM iodoacetamide and digested with sequencing grade trypsin at 37°C overnight. The digested peptides were separated by a 60 min gradient at a flow rate of 300 nl/min with Ultimate 3000 RSLCnano system. Mobile phase A is 0.1% formic acid, mobile phase B is 80% acetonitrile containing 0.1% formic acid. The analytical column was a homemade fused silica column (75 μm ID, 20 cm length) packed with C18 resin. Mass spectra were acquired using Orbitrap Eclipse Tribrid mass spectrometer. The resolution for full-scan spectra was 120,000, and the resolution for MS/MS spectra was 30,000. The cycle time was 3 s. Data were analysed by Proteome Discoverer 2.2.

### LC-MS analysis for small molecule identification

The purified KCC1 protein sample was initially buffer exchange into 200 mM ammonium acetate at pH 7.4 using a biopspin column (Bio-Rad) and resuspended in 100 μl of water with 0.1% formic acid to be analysed by LC-MS/MS. The separation was performed using an Ascentis Express C18 analytical column (0.3 × 150 mm, 2.7 μm) at 15 μl/min using the isocratic elution with following mobile phases, A = water with 0.1% formic acid and B = acetonitrile with 0.1% formic acid at 97:3 (v/v), respectively. The sample volume injected was 1 μl, and the total run time was 3 min. All measurements were performed using the Orbitrap Eclipse Tribrid mass spectrometer coupled with Ultimate 3000 binary pump. The MS was operated in negative polarity, and the ionization conditions were 275°C for capillary temperature (ion transfer tube), 20°C for vaporizer temperature and 3,500 V for spray voltage. Thermo Xcalibur software was used for data processing.

### NanoDSF measurements

All KCC variants were diluted to 0.2 mg/ml in protein buffer (20 mM HEPES pH 7.4, 150 mM NaCl, 0.04% digitonin). 10× nucleotide stock solutions were prepared in protein buffer. Protein and nucleotides were mixed and incubated on ice for 30 min. NT. Plex nanoDSF Grade High Sensitivity Capillaries (NanoTemper) were filled with 10-μl protein sample. Melting curves were determined in triplicates using Prometheus NT.48 by monitoring the intrinsic protein fluorescence signal as a measure of its folding state during a temperature ramp (1°C/min increase) from 20 to 95°C. Exemplary melting curves are shown in Appendix Fig S14. The melting temperature was determined by averaging the melting temperature of the triplicate measurements.

### Molecular dynamics simulations

The MD simulations were carried out using Desmond simulation package of Schrödinger LLC. The NPT ensemble with the temperature 300 K, and a pressure 1 bar was applied in all runs. The simulation length was 500 ns with a relaxation time 1 ps for the ligand ATP. The OPLS3e force field parameters were used in all simulations. The long-range electrostatic interactions were calculated using the particle mesh Ewald method (Toukmaji, 1996). The cut-off radius in Coulomb interactions was 9.0 Å. The water molecules were explicitly described using the simple point charge model (Zielkiewicz, 2005). The Martyna–Tuckerman–Klein chain coupling scheme (Martyna & Tuckerman, 1992) with a coupling constant of

2.0 ps was used for the pressure control and the Nosé–Hoover chain coupling scheme for the temperature control. Non-bonded forces were calculated using an r-RESPA integrator where the short-range forces were updated every step and the long-range forces were updated every three steps. The trajectories were saved at 200 ps intervals for analysis. The behaviour and interactions between the ligands and protein were analysed using the Simulation Interaction Diagram tool implemented in Desmond MD package. The stability of MD simulations was monitored by looking on the RMSD of the ligand and protein atom positions in time.

## Data availability

The cryo-EM density maps have been deposited into the Electron Microscopy Data Bank under the accession numbers EMD-10704 (https://www.emdataresource.org/EMD-10704), EMD-11799, (https://www.emdataresource.org/EMD-11799), EMD-11800 (https://www.emdataresource.org/EMD-11800), EMD-11801 (https://www.emdataresource.org/EMD-11801), EMD-11802 (https://www.emdataresource.org/EMD-11802), EMD-11803 https://www.emdataresource.org/EMD-11803) and EMD-12311 (https://www.emdataresource.org/EMD-12311).

The coordinates are deposited into the Protein Data Bank with accession numbers 6Y5V (https://www.rcsb.org/structure/6Y5V), 7AIN (https://www.rcsb.org/structure/unreleased/7AIN), 7AIO (https://www.rcsb.org/structure/unreleased/7AIO), 7AIP (https://www.rcsb.org/structure/unreleased/7AIP), 7AIQ (https://www.rcsb.org/structure/unreleased/7AIQ), 7AIR (https://www.rcsb.org/structure/unreleased/7AIR), 7NGB (https://www.rcsb.org/structure/unreleased/7NGB).

**Expanded View** for this article is available online.

## Acknowledgements
We thank the Dr. Dan Clare and Dr. Julika Radecke at eBIC (Didcot, United Kingdom), Dr. Christos Savva and Dr. TJ Ragan at Midlands Regional Cryo-EM Facility (Leicester, United Kingdom), Dr. Bilal Qureshi and Elizabeth Maclean at Oxford Particle Imaging Centre (Oxford, United Kingdom), Adam Costin and Dr. Errin Johnson at Dunn School Bioimaging Facility (Oxford, United Kingdom), Dr. Christian Wiesmann and Dr. Maryam Koshouei at Novartis (Basel, Switzerland), and Dr. Gregory Effantin and Dr. Michael Hons at ESRF (Grenoble, France) for their assistance with cryo-EM grid sample preparation, screening, and data collection. We also thank Dr. Ashley C. W. Pike, Dr. Jon Elkins, Dr Charline Giroud and Dr. Brian Marsden at Structural Genomics Consortium (Oxford, United Kingdom) for their assistance with model building, nanoDSF and cluster maintenance, respectively. This research was carried out with funding from the European Commission (Grant No. 115766). Oxford Particle Imaging Centre was funded by a Wellcome Trust JIF award (060208/Z/00/Z) and is supported by equipment grants from WT (093305/Z/10/Z). HT and CVR are funded an ERC Advanced Grant (695511). PI is funded by a Biomedical Research Grant from the Kidney Foundation of Canada.

## Author contributions
AFC, HM, GC, DW, TB, CPM, NAB-B and KLD designed and cloned the constructs. The protein was expressed by TB, HM, PA, GC, SMMM, GM, and the protein purification was done by HM, GC, RE. HM, RE and GC collected and processed the EM data. The model was built and refined by HM, GC, GR, FD, RE. Mass spectrometry experiments were conceived and performed by HT, XQ, FGA, IL, AJ and CVR. LET, APG, JLL, SS and PI designed and conducted Rb$^+$ uptake assays. RE, GC and KDB performed nanoDSF experiments. BT performed MD simulations. KLD, HM, GC, RE, HT wrote the manuscript. The manuscript was revised by KLD, HM, GC, RE, HT, PI and CVR.

## Conflict of interest
The authors declare that they have no conflict of interest.

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
