## [Review Process File · The EMBO Journal]

Phospho-regulation, nucleotide binding and ion access control in potassium-chloride cotransporters

Gamma Chi, Rebecca Ebenhoch, Henry Man, Haiping Tang, Laurence E. Tremblay, Gabriella Reggiano, Xingyu Qiu, Tina Bohstedt, Idir Liko, Fernando G. Almeida, Alexandre P. Garneau, Dong Wang, Gavin McKinley, Christophe Moreau, Kiran D. Bountra, Patrizia Abrusci, Shubhashish M. M. Mukhopadhyay, Alejandra Fernandez-Cid, Samira Slimani, Julie L. Lavoie, Nicola Burgess-Brown, Ben Tehan, Frank DiMaio, Ali Jazayeri, Paul Isenring, Carol V. Robinson, and Katharina L. Dürr
DOI: 10.15252/embj.2020107294

Corresponding author: Katharina Duerr (katharina.duerr@omass.com)

Review Timeline:

Submission Date:	12th Nov 20
Editorial Decision:	23rd Dec 20
Editorial Correspondence:	8th Jan 21
Revision Received:	27th Feb 21
Editorial Decision:	17th Mar 21
Revision Received:	29th Mar 21
Accepted:	11th Apr 21

Editor: Daniel Klimmeck

Transaction Report:

Dear Dr Duerr,

Thank you for the submission of your manuscript (EMBOJ-2020-107294) to The EMBO Journal. Please accept my sincere apologies for the unusual delay with the peer-review of your manuscript. Your study has been sent to three reviewers for evaluation, however one reviewer got much delayed and his-her report is still pending. We have received reports from the other two referees, which I enclose below, and decided to proceed with our decision based on these reports.

As you will see, the referees acknowledge the interest and novelty of your structural results. They also express a number of issues. In more detail, the referees request additional biochemical experimentation, complementary revision of data annotation and modelling as well as improved overall presentation of the findings and discussion, that would need to be conclusively addressed in their view to achieve the level of robustness and clarity needed for The EMBO Journal.

I judge the comments of the referees to be generally reasonable and given their overall interest, we are in principle happy to invite you to revise your manuscript to address the referees' comments, pending no technically overriding major concerns are raised by the remaining report of referee #1. I will share this report as soon as we will receive it.

Note that while per se well taken, more experiments on ATP hydrolysis (ref#2) and nucleotide binding of the transporters (ref#3, pt.3) are not strictly required for the current manuscript on our view.

Please let me know any time if you have additional questions or need further input on the referee comments.

Please see below for additional instructions for preparing your revised manuscript.

Thank you for the opportunity to consider your work for publication.
I look forward to your revision.

Kind regards,

Daniel Klimmeck

Daniel Klimmeck, PhD
Senior Editor
The EMBO Journal

Further information is available in our Guide For Authors:

The revision must be submitted online within 90 days; please click on the link below to submit the revision online before 23rd Mar 2021.

Link Not Available

Referee #2:

In this multi-disciplinary contribution by Chi et al., a few interesting findings, namely an additional nucleotide-binding site at the C-terminal domain, phosphorylation of regulatory sites and putative inhibition via N-terminus in potassium-coupled chloride transporters are presented. This is certainly important for the field and raises interesting points for further investigations on regulation of these transporters. The research is technically sound.

I do not have major concerns in the scientific part, but I found this manuscript difficult to read as it is overloaded with the figures, which in principle should make it easier to read, but here I got lost in all

main figures, extended figures and supplementary figures.

Another minor, but important point, do not use the term 'electron density' for Cryo-EM as it is (1) incorrect and (2) might be perceived as inappropriate by the EM community - for some reason they are extremely sensitive to this particular one. Also please avoid any expression of superiority like 'for the first time' etc. Also I found some inconsistencies in the way references to Figures are made, let's say first it refers to panel D in Figure 1 first, and only then to panel B, panel C and not to panel A at all.

In results section, p.4, it is mentioned that due to preferred particles orientation the data were processed to the reconstruction with the limited resolution - how limited? (give numbers).

on page 3, there is a typo - ioccurs, should be occurs

page 6 - instead of released structures, it's better to say - published structures

In thermostabilization results - I cannot analyse the supplementary Figure 4 F as it says - to be replaced in final version, so please update it. The so-called reversed thermostabilization effect for NKCC1 does not look like something functionally important and the difference is not that great (less than 1 degree between ADP and ATP).

In the discussion it is mentioned that the setup is far from ideal as it measures Rb⁺ uptake and not the efflux of K⁺/Cl⁻, so I wonder if the authors did try to measure the chloride activity with electrophysiology methods in oocytes - that would give more insight in the transport in my opinion.

On page 15, the suggestion is made that these proteins may exhibit ATP hydrolysis activity to drive conformational changes as seen in ABC-transporters, but this is easy to test - authors are encouraged to do a classical ATP-activity test with malachite green and also analyse whether there are structural elements in the vicinity of ATP-binding site, which can be used to translate the movements to trigger conformational changes.

My guess though, would be that this site is indeed rather regulatory, so could the authors please check if cyclic AMP can be accommodated in the binding site and perhaps also the thermostability with cAMP can be quickly evaluated.

On page 16 - please write 3DVA - variability analysis in full or introduce the abbreviation at the first occurrence.

I also missed the discussion on CTD alternative conformations in the discussion section - is this flexibility functional?

In figures with bars - please do not use full black as error bars become invisible.

In Figure S4 panel D and E, numbers are very hard to see.

Referee #3:

In this report by Chi and colleagues, the authors present cryo-EM reconstructions of mutant forms of KCC1 and KCC3, two members of the family of potassium-coupled chloride transporters. The structure of KCC3 was obtained using a construct (KCC3b-PM) in which three residues, S45, T940

and T997 were mutated to aspartate to mimic the phosphorylated state. KCC3b-PM displayed reduced activity compared to the wild-type transporter in Rb uptake assay. In the structure, a N-terminal domain was resolved at the interface between the TMD and the CTD in a position that would occlude access to the ion conduction pathway. To further examine the role of the N-terminal domain, the authors also determined the structure of a KCC1 mutant in which the first 19 amino acids were deleted (KCC1 Δ 19), which is proposed to represent a constitutively active form of the transporter. In the KCC1 Δ 19 structure, no density is present for the N-terminal domain and a non-protein density, suggested to be an ion, is resolved in the same place. Within a Rossmann-like fold the CTD of KCC1 Δ 19, an additional non-protein density is resolved that is proposed to represent an ATP or ADP. Notably, no nucleotide densities were resolved in the KCC3b-PM map. Molecular dynamics simulations suggest that ATP binds more strongly to KCC1 than to KCC3b, which may explain the differences resolved in the density maps.

While these structures and the proposed gating mechanisms represent important advancements in our understanding of these interesting transporters and warrant publication in EMBO Journal, the presentation of the manuscript needs to be improved prior to publication. For example, the font in the figures is not consistent (e.g. Supplementary Figure 10). Some figures lack proper labeling (e.g. Figure S5 A,B, Supplementary Figure 11) and some figures lack any reference in the article (Supplementary Figure 5, 6). It is also critical that prior to publication the authors clarify their proposal for the role of adenine nucleotides in the gating of the transporter, particularly if the authors believe that these molecules can regulate the activity of the transporters under physiological conditions.

Major points:

1. In the results section, the authors need to clarify precisely which constructs were used for structure determination, at what resolution the structures were determined and any potential artifacts present in the density maps of the different structures (i.e. preferred orientation of the particles).
2. Similarly, the authors should clearly state how the various constructs perform in their uptake assays. An explanation of any mutants that deviate from wild-type would be helpful. For example, the KCC1 Δ 19 mutant is described as constitutively active, yet has lower activity than activated wild-type KCC1. It is unclear if differences such as these are significant or correspond to differences in expression of the various constructs.
3. Using thermostability assays, the authors show that binding of adenine, but not guanine nucleotides enhances the stability of KCC1 and KCC3b. However, the authors do not directly explore the functional effects of nucleotide binding on transport activity. Moreover, the concentrations used (500 μ M) are far below those found in cells. The authors need to clarify that these experiments do not inform on the role of nucleotide binding in the gating of the transporter.
4. The authors state in the manuscript that: "The coexistence of two alternating CTD conformations in KCC1 Δ 19 is indicative of an increased flexibility around the TM-CTD-connecting scissor helices compared to KCC3b-PM. In line with this interpretation, we identified a bending region at residues 655-671 of KCC1 predicted by comparing the structures of state A and B (Figure S1 G)." However, the CTD of KCC3b-PM is poorly resolved in the density map which result from numerous potential conformations and would be consistent with analyses of other members of the family showing that the CTD is highly dynamic. The authors need to clarify the role of residues 655-671 in the flexibility of the CTD.
5. The authors state in the manuscript that: "Weaker density for the γ -phosphate indicates that a mixture of ADP- and ATP-bound particles may contribute the observed densities (Figure S5 C)." A clearer depiction of the relative intensity of each phosphate group should be presented to help readers. For KCC3b-PM, the authors state that "By contrast, no densities for ATP or ADP are visible

in the map of KCC3b-PM, which could be the result of small sequence differences in KCC3b." However, the local resolution of the potential nucleotide-binding site in KCC3b-PM seems to be very low. Is it possible that the density is too poorly resolved to be certain about the occupancy of the binding site?

6. During the review process, two other manuscripts describing structures of full-length KCC transporters were reported. It would be helpful if the authors were able to briefly comment in the discussion on any differences in the structures or their interpretation.

Minor points:

1. The manuscript states that: "To understand the structural basis of phospho-regulation in potassium-coupled chloride transporters, we used single-particle cryo-electron microscopy to study the full-length wild-type KCC3b transporter and mutants with replacements at the two canonical threonines, and a third KCC3-specific site (S96)[12]. We observed enhanced transport activity under isotonic and hypotonic conditions for the phospho-KO mutant (KCC3b-PKO: S45A/T940A/T997A ". It is not clear which positions are canonical and which is specific KCC3. How is position S45A related to S96?

2. Figure 1 B, the color of N-termini should be brighter.

3. Figure 1 C and D, for a better comparison the scale bars should be equal since both proteins reveal identical WT activity.

4. The manuscript states: "Due to preferred orientation of particles for KCC3b-WT and KCC3b-PKO, we determined structures of limited resolution for these constructs (Extended Figure 2 B, C and not shown), but obtained a 3D reconstruction at 3.2 Å for the phospho-inhibited KCC3b-PM variant (Figure 1 B)." What 'limited' resolution was reached? "Extended Figure 2 B, C" does not exist.

5. The manuscript states: "Full-length KCC1Δ19 and KCC3b-PM are domain-swapped dimers with each monomer in an inward-facing conformation." KCC1Δ19 is per definition not full-length.

6. Introduce a depiction of pore size with restriction site which indicates why the state is called inward-facing conformation.

7. The manuscript states: "Electrostatic surface representation (Figure 1 H) shows that one segment of the N-terminus in KCC3b-PM is wedged between CTD and TMD ...". Figure 1 H needs improvement, the details are not clear, the pore region is not clearly depicted.

8. The manuscript states: "for cations coordinated by the highly conserved residues of the cation binding site ...". Which of the residues are highly conserved?

9. The manuscript states: "In agreement with the secondary structure predictions, the second half of the resolved N-terminal domain of KCC3b-PM". Typo.

10. The manuscript states: "The most N-terminal residues in KCC3b-PM (M101-G111) lack secondary structure element ". What does the term 'the most N-terminal' mean?

11. The manuscript states: "To gain further insights into the functional significance of the N-terminal interactions observed in the KCC3b-PM structure, we deleted the respective 20 amino acid stretch mentioned before in various KCC constructs and determined how Rb⁺ uptake activities in

Xenopus oocytes are affected." It is not clear which residues are deleted and how this deletion is corresponding to KCCΔ19 mutant. Authors should indicate the position of the exact residues for each mutation.

12. Figure S2 H is very crowded and it is hard to estimate which positions are important.

13. The manuscript states: "These unique phosphorylation sites in KCC1 are located near a conserved structural motif closely resembling the ATP-binding motif in bacterial universal stress proteins (USPs)." What is the kind of resembling (structure/sequence?) of the ATP-binding motif in bacteria?

14. The manuscript states: "Similar to the ATP binding mode in HeTeaD (Figure S5 E, Supplementary Figure 13 C), the purine base is stabilized by polar interactions with the backbone peptide to V730 in the β2-strand of the CTD." Authors should explain how HeTeaD is related to KCC1 and KCC3? What is the sequence identity?

15. The manuscript states: "Notably, we observed phosphorylation for T727 in KCC3b by proteomics analysis (Supplementary Figure 9 D), but not for the corresponding T713 in KCC1 (Supplementary Figure 8 G)." Supplementary Figure 9 D does not contain related data.

16. The manuscript states: "One of the most significant differences in the sequence that may explain the lack of ATP in the KCC3 map is H722, whose side chain is only seen to be interacting with the nucleotide greater than one third of the time for two of the three simulations conducted (Supplementary Figure 12 C-F)." How do authors explain different outcomes for the same MD on KCC3-CTD? Based on the Supplementary Figure 12 E and F, it seems like the coordination via Lys723 is essential for phosphate coordination even in a stronger way than H722. Can authors comment on this finding?

Dear Dr Duerr,

This is to share the delayed report by reviewer #1 for your manuscript EMBOJ-2020-107294.

Please consider these comments for your revised manuscript.

Please let me know if you have any question related.

Kind regards,

Daniel Klimmeck

EMBOJ-2020-107294;

Referee #1:

Chi, Ebenhoch, Mann, Tang et al. & Dürr report a very exciting study of KCC1/KCC3 with implications also for other members of the CCC/SLC12 family, and representing probably the most enlightening new structural study of the CCC family so far.

Key findings include their description of i) rotations of the cytoplasmic domains relative to the TM axis and in relation to functional transition, ii) autoregulatory function of phosphosites and N-terminal motifs, and iii) ATP/ADP binding at the cytoplasmic domains. This is a careful and insightful report and is in no way harmed by very recent publications/preprints on KCC1/2/3/4 structures (<https://advances.sciencemag.org/content/6/50/eabc5883>), which in comparison to this study are rather superficial.

Structural and functional characterizations are generally reported well with only few revisions requested (see below). The report is conclusive, justified and well represented at the current level with important findings for a broad readership. Further functional analysis by mutational studies and cell biological function should become interesting follow-up studies.

Major points

The cryo-EM data processing (suppl. methods) is confusing and must be made clearer.

Why do they start with C2 symmetry imposed and later expand into C1 processing instead of starting in C1 and later investigate C2? This must be explained further.

Terminology/accounts on particle numbers should also be revisited it seems - the particle set must be the same, only internal symmetry operations change

It would be interesting to address briefly the structural role of ADP/ATP binding in relation to functional transitions

Minor points:

Table 1 - pixel size should have a unit

Supplementary methods need to be updated with actual table and figure numbers not just #####. A SI table ##### is referred to - probably Table1?

Response-to-referees (EMBOJ-2020-107294R)

**Phospho-regulation, nucleotide binding and ion access control in
K-Cl cotransporters**

Corresponding author: Katharina L. Dürr

We thank the referees for a very thorough evaluation. We have addressed all raised points, as detailed below.

Referee #1:

Chi, Ebenhoch, Mann, Tang et al. & Dürr report a very exciting study of KCC1/KCC3 with implications also for other members of the CCC/SLC12 family, and representing probably the most enlightening new structural study of the CCC family so far.

Key findings include their description of i) rotations of the cytoplasmic domains relative to the TM axis and in relation to functional transition, ii) autoregulatory function of phosphosites and N-terminal motifs, and iii) ATP/ADP binding at the cytoplasmic domains. This is a careful and insightful report and is in no way harmed by very recent publications/preprints on KCC1/2/3/4 structures (<https://advances.sciencemag.org/content/6/50/eabc5883>), which in comparison to this study are rather superficial.

Structural and functional characterizations are generally reported well with only few revisions requested (see below). The report is conclusive, justified and well represented at the current level with important findings for a broad readership. Further functional analysis by mutational studies and cell biological function should become interesting follow-up studies.

Thank you, we appreciate the positive feedback!

Major points

The cryo-EM data processing (suppl. methods) is confusing and must be made clearer.

Why do they start with C2 symmetry imposed and later expand into C1 processing instead of starting in C1 and later investigate C2? This must be explained further. Terminology/accounts on particle numbers should also be revisited it seems - the particle set must be the same, only internal symmetry operations change

We initially processed the datasets with no imposed symmetry (i.e. C1 symmetry; 3.7 Å) and then found that the two subunits were nearly identical, leading us to reprocess in C2 symmetry for better resolutions. On visual examination, we found that the resolutions for the datasets were improved, e.g. side chain densities becoming clearer. ATP and its surrounding residues particularly benefited from this reprocessing, so we decided to deposit the map processed in C2 symmetry.

Additionally, the minor species (subclasses of KCC3b-PM and KCC1-Δ19 datasets) were very difficult to isolate with heterogeneous refinement (Cryosparc 2.14) and static 3D classification (Relion 3.1). This difficulty was probably due to 1) the difference between the subclasses that were very small compared to the overall structures, and 2) the pseudo-C2 symmetry of these minor species that often caused the particles to be assigned wrong orientations. Processing in C2 symmetry followed by symmetry expansion and 3D variability analysis allowed us to isolate these particles much better, and this is another reason why we opted to use the C2-imposed map as the main consensus one.

It would be interesting to address briefly the structural role of ADP/ATP binding in relation to functional transitions

At this point, we are not sure of the exact role of ATP in functional transitions and conclusive answers will require additional experiments (see also response to reviewer #2), but we added a few lines to the discussion on page 16/17 with suggestions of possible roles based on observations from the literature, which hint that accessibility of inhibitor-binding sites in the ECD could be state-dependent and positively affected by the presence of nucleotides. Adenine nucleotides were shown to increase 3H bumetanide binding in NKCC1 and this could be due to a shift in the conformational equilibrium in favour to the outward-open state with a high affinity binding site in the ECD, assuming a similar binding mechanism as recently proposed by the structure of KCC1 in the outward-open, VU0463271 inhibitor-bound state.

Minor points:

Table 1 - pixel size should have a unit

The unit of the pixel size has been added to Table 1.

Supplementary methods need to be updated with actual table and figure numbers not just #####. A SI table ##### is referred to - probably Table1

The Appendix Method file has been updated with the actual Figure and Table numbers, i.e., Table 1 and Appendix Figures S3 and S4.

Referee #2:

In this multi-disciplinary contribution by Chi et al., a few interesting findings, namely an additional nucleotide-binding site at the C-terminal domain, phosphorylation of regulatory sites and putative inhibition via N-terminus in potassium-coupled chloride transporters are presented. This is certainly important for the field and raises interesting points for further investigations on regulation of these transporters. The research is technically sound.

I do not have major concerns in the scientific part, but I found this manuscript difficult to read as it is overloaded with the figures, which in principle should make it easier to read, but here I got lost in all main figures, extended figures and supplementary figures.

Thank you for the comment. We have reorganized the manuscript and rearranged the figures to improve readability. For example, former panel H from Figure 1 is now part of Figure 2, and Figure 5 has been split into two Figures (5 and 6) with content-related Extended View Figures. New panels have been added to Extended view Figure 1 and Figure 2 to address some of the points raised by other reviewers. The supplementary figures (Appendix Figures S1-S14) have also been rearranged to fit better the order in which they are referred to in the manuscript.

Another minor, but important point, do not use the term 'electron density' for Cryo-EM as it is (1) incorrect and (2) might be perceived as inappropriate by the EM community - for some reason they are extremely sensitive to this particular one.

We have removed the term “electron density map” and replaced it by “density map” or “electron scatter map”.

Also please avoid any expression of superiority like 'for the first time' etc.

We have addressed this issue and are now avoiding expressions of novelty or superiority throughout the manuscript.

Also I found some inconsistencies in the way references to Figures are made, let's say first it refers to panel D in Figure 1 first, and only then to panel B, panel C and not to panel A at all.

Thanks for raising this point – we have adjusted the text to include references to these Figure panels in the right order.

In results section, p.4, it is mentioned that due to preferred particles orientation the data were processed to the reconstruction with the limited resolution - how limited? (give numbers).

We have added a number for the resolution we obtained for the KCC3b-PKO data (6.5 Å).

on page 3, there is a typo - ioccurs, should be occurs

Correction made.

page 6 - instead of released structures, it's better to say - published structures

Done.

In thermostabilization results - I cannot analyse the supplementary Figure 4 F as it says - to be replaced in final version, so please update it.

This panel in the Appendix Figure S14 has been updated and the comment “to be replace in final version” removed.

The so-called reversed thermostabilization effect for NKCC1 does not look like something functionally important and the difference is not that great (less than 1 degree between ADP and ATP).

We agree and have therefore rephrased the sentence, stating only that NKCC1 shows a lower degree of thermostabilisation.

In the discussion it is mentioned that the setup is far from ideal as it measures Rb⁺ uptake and not the efflux of K⁺/Cl⁻, so I wonder if the authors did try to measure the chloride activity with electrophysiology methods in oocytes - that would give more insight in the transport in my opinion.

Thank you for the suggestion. For the current study, the Rb uptake method was selected because it is robust and the one most commonly used to characterize the KCCs. The proposed method of using chloride-conducting channels (GABA or Glycine receptors) to monitor changes in the chloride reversal potential caused by the KCC-mediated decrease of the intracellular chloride concentration are also common in the field, but require co-expression with these channels and are hence indirect

in nature. Direct measurement of chloride extrusion by KCCs by electrophysiology is not feasible due to the electroneutral transport mode of these transporters.

On page 15, the suggestion is made that these proteins may exhibit ATP hydrolysis activity to drive conformational changes as seen in ABC-transporters, but this is easy to test - authors are encouraged to do a classical ATP-activity test with malachite green and also analyse whether there are structural elements in the vicinity of ATP-binding site, which can be used to translate the movements to trigger conformational changes.

Of notice, we have added a section on page 12 highlighting possible structural determinants in the vicinity of the ATP binding site (eg. residues which could serve as a “general base” for activating a water molecule or for Mg^{2+} coordination) along with a sequence alignment to illustrate conservation across the SLC12 family (Figure EV5 B). Thank you for the suggestion of testing ATP hydrolysis by the classical test with malachite green. While it is true that such assays are easy to perform, conclusive answers require the inclusion of several controls, like mutants in the ATP-coordinating residues or the structural elements which could play a role in hydrolysis or triggering conformational changes. Such controls are important because otherwise we cannot rule out the possibility that ATP is hydrolysed by HSP70 or HSP90 proteins, which are known contaminants in the BacMam/HEK293 expression system, or kinases which might be present as low-level contaminants. The inclusion of HSP/kinase inhibitors alleviate this problem to a certain degree, but full mechanistic insights into the potential role of ATP hydrolysis and associated conformational changes require the generation and purification of a series of mutants, and ideally also their characterization in a liposomal transport assay. We would appreciate if we could complete the revision in the suggested 90 days and present the results from these comprehensive experiments in a separate, future study.

My guess though, would be that this site is indeed rather regulatory, so could the authors please check if cyclic AMP can be accommodated in the binding site and perhaps also the thermostability

Figure 1. cAMP fitted into density map of KCC1. Compare to EV Figure 5A with ATP fitted instead.

with cAMP can be quickly evaluated.

Thanks for the suggestion. cAMP would indeed fit into the pocket as well (see Figure 1 of this response), but it does not fully account for the observed density in the map of KCC1, whereas the beta and gamma-phosphates present in ATP nicely fit into the densities (EV Figure 5 A).

We also tested cAMP (500 μ M) experimentally in nanoDSF experiments and did not see any significant effect on the melting behaviour of KCC1 (Figure included). In vivo, concentrations of cAMP are typically 10-fold lower (below 50 μ M). It is thus unlikely that cAMP binds to the pocket under physiological conditions, all the more so that it would have to compete against ATP, which is present in low mM concentrations in the cell.

Figure 2. cAMP effect on melting behaviour of KCC1 in a nanoDSF experiment.

On page 16 - please write 3DVA - variability analysis in full or introduce the abbreviation at the first occurrence.

We have introduced the abbreviation “3DVA” on page 9.

I also missed the discussion on CTD alternative conformations in the discussion section - is this flexibility functional?

This is a difficult question we unfortunately cannot fully answer with the current results from the “in vitro” experiments. However, we have expanded on the discussion regarding this issue and included structural data from related studies by other groups, data which were released while the current work was under review (Figure 3 F-H). These structures support the alternative conformations suggested by 3DVA and HDX-MS data.

In figures with bars - please do not use full black as error bars become invisible.

We have replaced black bars in the thermostability bar chart by dark grey bars (Fig 6 B).

In Figure S4 panel D and E, numbers are very hard to see.

Thank you, we have replaced the numbers in these panels by a larger font size (now Fig EV6 D, E) for improved readability.

Referee #3:

In this report by Chi and colleagues, the authors present cryo-EM reconstructions of mutant forms of KCC1 and KCC3, two members of the family of potassium-coupled chloride transporters. The structure of KCC3 was obtained using a construct (KCC3b-PM) in which three residues, S45, T940 and T997 were mutated to aspartate to mimic the phosphorylated state. KCC3b-PM displayed reduced activity compared to the wild-type transporter in Rb uptake assay. In the structure, a N-terminal domain was resolved at the interface between the TMD and the CTD in a position that would occlude access to the ion conduction pathway. To further examine the role of the N-terminal domain, the authors also determined the structure of a KCC1 mutant in which the first 19 amino acids were deleted (KCC1 Δ 19), which is proposed to represent a constitutively active form of the transporter. In the KCC1 Δ 19 structure, no density is present for the N-terminal domain and a non-protein density, suggested to be an ion, is resolved in the same place. Within a Rossmann-like fold the CTD of KCC1 Δ 19, an additional non-protein density is resolved that is proposed to represent an ATP or ADP. Notably, no nucleotide densities were resolved in the KCC3b-PM map. Molecular dynamics simulations suggest that ATP binds more strongly to KCC1 than to KCC3b, which may explain the differences resolved in the density maps.

While these structures and the proposed gating mechanisms represent important advancements in our understanding of these interesting transporters and warrant publication in EMBO Journal, the presentation of the manuscript needs to be improved prior to publication. For example, the font in the figures is not consistent (e.g. Supplementary Figure 10). some figures lack proper labeling (e.g. Figure S5 A,B, Supplementary Figure 11) and Some figures lack any reference in the article (Supplementary Figure 5, 6).

The font in Supplementary Figure 10 (now Appendix Figure S11) has been changed to be consistent across all four panels. Labels of former Figure S5 A, B (now Figure 5 A, B) and former Supplementary Figure 11 (now Appendix Figure S5) have been improved. Former Supplementary Figures 5 and 6 with the EM workflows (now Appendix Figures S3 & S4) are referenced on pages 4 and 5, respectively.

It is also critical that prior to publication the authors clarify their proposal for the role of adenine nucleotides in the gating of the transporter, particularly if the authors believe that these molecules can regulate the activity of the transporters under physiological conditions.

In the discussion, we have proposed a few hypotheses for the potential role of nucleotide binding in regulating transport activity and referred to reports in the literature which support these ideas (page 16-17). The current experimental data unfortunately does not allow us to understand the physiological relevance or role of ATP or ADP binding. Addressing this question in the Rb uptake assay in oocytes is not feasible, because it is difficult to manipulate intracellular nucleotide concentrations in living cells. Rb flux measurements with mutants that selectively affect nucleotide binding would be very insightful, but it takes time to generate and assess nucleotide binding for a range of mutants in vitro and subsequently characterize these further in vivo. In vitro transport assays with liposome-reconstituted KCCs are also a possible approach to understand the role of nucleotide binding for transport activity. See also response to reviewer 1.

Major points:

1. In the results section, the authors need to clarify precisely which constructs were used for structure determination, at what resolution the structures were determined and any potential

artifacts present in the density maps of the different structures (i.e. preferred orientation of the particles).

We added the construct details and map resolution for each dataset to Table 1 and clarified that only the dataset of the KCC3-PKO mutant was severely affected by preferred orientation. Therefore, we do not discuss the resulting 6.5 Å structure at all and focus on characterization of this mutant by HDX-MS instead. The KCC3b-WT maps from datasets in LMNG or MSP nanodisc were only moderately affected by preferred orientation. Any artifacts in the map would not affect our major conclusions, because we only discuss the lack of density for the helical portion of the N-terminal peptide for these structures (Figure EV2 B, C) and do not discuss the positioning of individual residues, ligands or ions, which might be affected by such map artefacts. Angular distributions for the two structures we discussed in detail in this manuscript are shown in Appendix Figures S3 G and S4 G, respectively.

2. Similarly, the authors should clearly state how the various constructs perform in their uptake assays. An explanation of any mutants that deviate from wild-type would be helpful. For example, the KCC1Δ19 mutant is described as constitutively active, yet has lower activity than activated wild-type KCC1. It is unclear if differences such as these are significant or correspond to differences in expression of the various constructs.

We have expanded on this section and describe in more detail how the different construct modifications affect Rb uptake under isotonic and hypotonic conditions. We also rephrased the description of the phenotype of KCC1Δ19, i.e., we put more emphasis on the similar uptake levels under hypotonic and isotonic activity observed for this mutant. Importantly, the data show that unlike the WT construct, the mutant is activated even under isotonic conditions, and not further stimulated by hypotonicity. The overall lower uptake activity of KCC1Δ19 compared to wild-type KCC1 after hypotonic stimulation could indeed be a consequence of lower expression or plasma membrane targeting. However, the fact that we always show isotonic and hypotonic conditions for each construct helps because the ratio of these activities is the most interesting aspect we are discussing. Therefore, we believe that reduced expression or surface delivery for the mutants would not change our overall conclusions.

3. Using thermostability assays, the authors show that binding of adenine, but not guanine nucleotides enhances the stability of KCC1 and KCC3b. However, the authors do not directly explore the functional effects of nucleotide binding on transport activity. Moreover, the concentrations used (500 μM) are far below those found in cells. The authors need to clarify that these experiments do not inform on the role of nucleotide binding in the gating of the transporter.

We have tested ATP and ADP on KCC1 in thermal shift experiments over a physiological concentration range: up to 27 mM for ATP (see included data in this response) and 3 mM for ADP (included here and Appendix Figure S13 A) respectively. For cAMP, AMP and GTP we only included a single-point at 500 μM, i.e., at a concentration that is above or within the physiological range, but did not pursue further, in particular for GTP that would not fit into the pocket well.

As mentioned, it is true that in the current study we did not include experiments that could have informed on the role of nucleotide binding for gating of the transporter but we are now better explaining why on page 15 of the revised manuscript. We mention for instance that “There are no ideal methods to alter intracellular nucleotide concentrations predictably in living oocytes... *In vitro* transport assays with KCCs reconstituted into proteoliposomes could be an elegant way to address

this important question in future studies, all the more so that the feasibility of such assays has been recently demonstrated for KCC1 [50].”

Figure 3. nanoDSF melting curves of KCC1 in presence of different ATP and ADP concentrations.

4. The authors state in the manuscript that: "The coexistence of two alternating CTD conformations in KCC1 Δ 19 is indicative of an increased flexibility around the TM-CTD-connecting scissor helices compared to KCC3b-PM. In line with this interpretation, we identified a bending region at residues 655-671 of KCC1 predicted by comparing the structures of state A and B (Figure S1 G)." However, the CTD of KCC3b-PM is poorly resolved in the density map which result from numerous potential conformations and would be consistent with analyses of other members of the family showing that the CTD is highly dynamic. The authors need to clarify the role of residues 655-671 in the flexibility of the CTD.

Based on our Dyndom analysis shown in Figure 1 H, residues 655-671 of KCC1 are the hinge point for the conformational transition between the two CTD conformations shown in Figure 1 E and F. In or HDX-MS data for KCC3 (Figure 3A), the corresponding region (residues 682-691) is also flexible. We therefore concluded that in both family members this region of the transporter (called the scissor helix) is a flexible connection between TMD and CTD. A phosphorylation site in the scissor helix of KCC3b (S685 in Figure 4 A) could play a conformational role in this region.

5. The authors state in the manuscript that: "Weaker density for the γ -phosphate indicates that a mixture of ADP- and ATP-bound particles may contribute the observed densities (Figure S5 C)." A clearer depiction of the relative intensity of each phosphate group should be presented to help readers. For KCC3b-PM, the authors state that "By contrast, no densities for ATP or ADP are visible in the map of KCC3b-PM, which could be the result of small sequence differences in KCC3b." However, the local resolution of the potential nucleotide-binding site in KCC3b-PM seems to be very low. Is it possible that the density is too poorly resolved to be certain about the occupancy of the binding site?

We have enlarged the Figure showing the density for the putative nucleotide in Fig EV 5 A and labelled the beta and gamma-phosphates for better clarity. The cryo-EM data alone are indeed not sufficient to exclude that ATP/ADP is bound to KCC3b-PM. We have thus included the MD and thermal shift data to explore this question further.

6. During the review process, two other manuscripts describing structures of full-length KCC transporters were reported. It would be helpful if the authors were able to briefly comment in the discussion on any differences in the structures or their interpretation.

The reported KCC2-4 structures are similar and exhibit the N-terminal autoinhibition described in our study. However, none of these studies did mention nucleotide binding to the CTD. Notable differences in the conformation of the alpha8 helix in the CTD of the reported KCC3b structure (Cell Res 2020) exist, which are shown in a new panel F added to Figure 3. In the published KCC3a structure (Sci Adv 2020) the respective region has been omitted from model building. We also show the alpha8 conformation from the CeKCC1 CTD X-ray structure in Figure 3 G for comparison, and added a few sentences summarizing these differences and explain how the variability in these particular regions actually agrees well with our 3DVA and HDX-MS results (page 10 of the revised Manuscript).

Minor points:

1. The manuscript states that: "To understand the structural basis of phospho-regulation in potassium-coupled chloride transporters, we used single-particle cryo-electron microscopy to study the full-length wild-type KCC3b transporter and mutants with replacements at the two canonical threonines, and a third KCC3-specific site (S96)[12]. We observed enhanced transport activity under isotonic and hypotonic conditions for the phospho-KO mutant (KCC3b-PKO: S45A/T940A/T997A)". It is not clear which positions are canonical and which is specific KCC3. How is position S45A related to S96?

We have added a sentence to explain that positions T940 and T997 are the two canonical phosphorylation sites conserved within the K branch of the SLC12 family and that the S45A mutant (B-isoform labelling) has been shown to play a role for full activation of KCC3 by swelling. S45 corresponds to S96 in KCC3 isoform A. To avoid confusion, we now state in the MS that we use the numbering based on the B-isoform.

2. Figure 1 B, the color of N-termini should be brighter.

Done.

3. Figure 1 C and D, for a better comparison the scale bars should be equal since both proteins reveal identical WT activity.

Done.

4. The manuscript states: "Due to preferred orientation of particles for KCC3b-WT and KCC3b- PKO, we determined structures of limited resolution for these constructs (Extended Figure 2 B, C and not shown), but obtained a 3D reconstruction at 3.2 Å for the phospho-inhibited KCC3b-PM variant (Figure 1 B). " What 'limited' resolution was reached? "Extended Figure 2 B, C" does not exist.

The achieved resolution (6.5 Å for KCC3b-PKO) was added. EV Fig 2B, C are showing the maps of KCC3-WT structures (in detergent: Fig EV2B, nanodisc: Fig EV2C) around the N-terminal region to highlight the differences compared to the KCC3b-PM structures (Fig EV2A,D) despite the lower resolution.

5. The manuscript states: "Full-length KCC1Δ19 and KCC3b-PM are domain-swapped dimers with each monomer in an inward-facing conformation. " KCC1Δ19 is per definition not full-length.

We have removed the term full-length.

6. Introduce a depiction of pore size with restriction site which indicates why the state is called inward-facing conformation.

Figures EV1 B, C with surface representations of the monomeric TM regions have been added to illustrate the inward facing conformations of KCC1 and KCC3. Such representations are common for solute carriers and usually considered sufficient to demonstrate the inward-facing state. Running the coordinates through pore-walker (<https://www.ebi.ac.uk/thornton-srv/software/PoreWalker/>) did not produce any meaningful outputs. As such, we haven't included a plot for the pore radius which is common practice for ion channel structures.

7. The manuscript states: "Electrostatic surface representation (Figure 1 H) shows that one segment of the N-terminus in KCC3b-PM is wedged between CTD and TMD ...". Figure 1 H needs improvement, the details are not clear, the pore region is not clearly depicted.

Figure 1 H has been moved to Figure 2 where panels B and C now show the surface representation of KCC3B-PM in the absence and presence of the N-terminus to illustrate that this protein domain obstructs the intracellular vestibule.

8. The manuscript states: "for cations coordinated by the highly conserved residues of the cation binding site ...". Which of the residues are highly conserved?

The four conserved residues in KCC3b shown in Figure EV2E are now explicitly mentioned in the text on page 7 now: Y232, I146, P443 and T446.

9. The manuscript states: "In agreement with the secondary structure predictions, the second half of the resolved N- terminal domain of KC3b-PM". Typo.

Typo has been corrected.

10. The manuscript states: "The most N-terminal residues in KCC3b-PM (M101-G111) lack secondary structure element ". What does the term 'the most N-terminal' mean?

We are now actually indicating the residues in brackets. The sentence has also been rephrased.

11. The manuscript states: "To gain further insights into the functional significance of the N-terminal interactions observed in the KCC3b-PM structure, we deleted the respective 20 amino acid stretch mentioned before in various KCC constructs and determined how Rb⁺ uptake activities in *Xenopus* oocytes are affected. " It is not clear which residues are deleted and how this deletion is corresponding to KCCΔ19 mutant. Authors should indicate the position of the exact residues for each mutation.

We have added the residue numbers for the deletions in the Figure legend of Figure 2 I, J to make it clear which residues were deleted.

12. Figure S2 H is very crowded and it is hard to estimate which positions are important.

The most important residues for the CTD/NT interaction in KCC3 have now been highlighted by black boxes.

13. The manuscript states: "These unique phosphorylation sites in KCC1 are located near a conserved structural motif closely resembling the ATP-binding motif in bacterial universal stress proteins (USPs). " What is the kind of resembling (structure/sequence?) of the ATP-binding motif in bacteria?

See next query and Ref 45. ATP binding motifs are rather diverse and difficult to identify by sequence analysis, which explains why the binding pocket has not been identified prior to the current structural study.

14. The manuscript states: "Similar to the ATP binding mode in HeTeaD (Figure S5 E, Supplementary Figure 13 C), the purine base is stabilized by polar interactions with the backbone peptide to V730 in the β2-strand of the CTD. " Authors should explain how HeTeaD is related to KCC1 and KCC3? What is the sequence identity?

A sentence regarding this issue has been added on page 11. We mention that the sequence identity with *HeTeaD* is very low, precluding traditional sequence alignments. We therefore cannot give a percentage in sequence identity and the similarity was only discovered by searches looking for

structural similarities in other deposited structures using the DALI server: <http://ekhidna2.biocenter.helsinki.fi/dali/>
 Sequence alignment is only possible when using structural constraints, for example with PROMALS (<http://prodata.swmed.edu/promals/promals.php>). Result from such a structural alignment is included here as an example:

Supplementary Figure XY - PROMALS3D sequence alignment between *HsKCC1* and *HeTeaD* using the KCC1Δ19 (7AIRQ) and 3HGM as input files for structural constraints (weight factor 1.5). Secondary structure elements are represented as pink α -helices (h) and blue β -strands (e). Consensus symbols: conserved amino acids are in bold and uppercase letters; aliphatic (I, V, L): l; aromatic (Y, H, W, F): @; hydrophobic (W, F, Y, M, L, I, V, A, C, T, H): h; alcohol (S, T): o; polar residues (D, E, H, K, N, Q, R, S, T): p; tiny (A, G, C, S): t; small (A, G, C, S, V, N, D, T, P): s; bulky residues (E, F, I, K, L, M, Q, R, W, Y): b; positively charged (K, R, H): +; negatively charged (D, E): -; charged (D, E, K, R, H): c.

Figure 4. PROMALS3D sequence alignment between HsKCC1 and HeTeaD using the KCC1Δ19 coordinates (7AIRQ).

15. The manuscript states: "Notably, we observed phosphorylation for T727 in KCC3b by proteomics analysis (Supplementary Figure 9 D), but not for the corresponding T713 in KCC1 (Supplementary Figure 8 G)." Supplementary Figure 9 D does not contain related data.

There was a mistake in the Figure legend for Supplementary Figure 9 D (now Appendix Figure S6). In fact, the peptide starts at residue Leu-725 and hence shows that that T727 is phosphorylated (position three in the fragmented peptide). This error has been corrected in the revised Appendix Figure S6.

16. The manuscript states: "One of the most significant differences in the sequence that may explain

the lack of ATP in the KCC3 map is H722, whose side chain is only seen to be interacting with the nucleotide greater than one third of the time for two of the three simulations conducted (Supplementary Figure 12 C-F). " How do authors explain different outcomes for the same MD on KCC3-CTD? Based on the Supplementary Figure 12 E and F, it seems like the coordination via Lys723 is essential for phosphate coordination even in a stronger way than H722. Can authors comment on this finding?

In each MD run, the velocities of each atom in the simulation are assigned randomly from a Maxwell-Boltzmann distribution representing a certain temperature (in this case 300 K). Different outcomes for ligand/protein interactions are thus possible, even if all other parameters are kept constant. This is because a certain atom with a key role in the interaction might have a different starting velocity in run 1 compared to run 2 and 3, which can affect the result of the simulation. For this reason, we carried out multiple runs for each structure, and the runs with KCC3 has consistently resulted in output files with a lower number of stabilizing interactions to ATP.

The reviewer is correct that K721 in KCC3 is engaging into a strong interaction with oxygen atoms of the β - and γ -phosphate groups. This residue corresponds to K707 in KCC1 and is thus conserved between the two family members (see also Fig EV5B). The stabilising interaction of K721 shown in Appendix Figure S13 D is therefore not surprising, and we presume that the smaller pocket and changed geometry caused by a larger hydrophobic residue (L784) is responsible for the observation that this interaction is not present in all MD runs performed for KCC3. In the section where we discuss H722, we explicitly focus on sequence differences between KCC1 and KCC3. As for the shared residues, their roles were discussed earlier in the manuscript when the corresponding lysine interaction of KCC1 was described (page 11).

Dear Dr Dürr,

Thank you for submitting your revised manuscript (EMBOJ-2020-107294R) to The EMBO Journal. Your amended study was sent back to the referees for re-evaluation, and we have received comments from referees #2 and #3, which I enclose below. Please note that while referee #1 was at this time not able to reassess the work, we have editorially evaluated your response to his/her concerns and found them to be convincingly addressed. As you will see, the other referees stated that their issues have been comprehensively resolved and are now broadly in favour of publication, pending minor revision.

Thus, we are pleased to inform you that your manuscript has been accepted in principle for publication in The EMBO Journal.

Please consider the remaining minor issues stated by the referees carefully and address them by adding complementary annotation and literature.

In addition, we need you to take care of a number of points related to formatting and data representation as detailed below, which should be addressed at re-submission.

Please contact me at any time if you have additional questions related to below points.

Thank you for giving us the chance to consider your manuscript for The EMBO Journal. I look forward to your final revision.

Again, please contact me at any time if you need any help or have further questions.

Kind regards,

Daniel Klimmeck

- >> Please add a 'Conflict of Interest' section after the Acknowledgments.
- >> Add maximally five keywords to your manuscript.
- >> Recheck callouts and their correct order in the main text for Fig. 5D and EV4(5,6).
- >> Please specify author contributions for C.M. and P.A. .

- >> Complement your data access section with the weblinks for the data sets.
- >> Rename the 'Methods' section to 'Material and Methods'.
- >> Recheck if the Zhao et al. bioRxiv entries was published in the meantime and update the reference in case.
- >> State redisplay of Fig3B,C in the legends of FigEV5B,C.
- >> Please re-format all EV files in numerical order - not skipping over a number. Update all callouts and nomenclature accordingly.
- >> Update the Appendix ToC, adding the Appendix figure numbers.
- >> Remove Movie legends from the MS. The names need correction to Movie EV#. Please zip together the corresponding movie file and movie legend in .docx format.
- >> Adjust the reference format to EMBO Journal style, limiting to 10 authors et al. .
- >> As we do not offer the option to state, 'data not shown', please amend the data referred to at p.5, first line, or remove the statement.
- >> Please consider additional changes and comments from our production team as indicated by the .doc file enclosed and leave changes in track mode.

Please remember: Digital image enhancement is acceptable practice, as long as it accurately

represents the original data and conforms to community standards. If a figure has been subjected to significant electronic manipulation, this must be noted in the figure legend or in the 'Materials and Methods' section. The editors reserve the right to request original versions of figures and the original images that were used to assemble the figure.

The revision must be submitted online within 90 days; please click on the link below to submit the revision online before 15th Jun 2021.

Referee #2:

I thank the authors for the substantial revision and that they sufficiently addressed the points I raised.

Three very minor things - in the methods section, one reference is missing: Baculoviruses for these constructs were generated following the standard protocol outlined in [REF].

In Table 1, one of the microscopes is Titan Krios, (ESRF, UK), but if it is ESRF, that is in France, unless there is another ESRF somewhere in UK, but then it is super confusing

In Figure 2, panel H is cropped.

Overall, I recommend this manuscript for publication in the EMBO Journal

Referee #3:

The authors have addressed all of the raised minor and major points, which greatly improved the structure of the paper and makes it easier to understand. In principle, the manuscript is ready for publication. However, the authors should double-check their figures and tables which still contain some minor flaws (e.g. inconsistent formatting, missing text etc.). Furthermore, the authors need to precisely indicate whether they are referring to wildtype or mutant proteins in their text.

Point-by-point response to the referees' comments (EMBOJ-2020-107294R)

>> Please add a 'Conflict of Interest' section after the Acknowledgments.

Done.

>> Add maximally five keywords to your manuscript.

5 keywords were added after the abstract in the manuscript file.

>> Recheck callouts and their correct order in the main text for Fig. 5D and EV4(5,6).

Done.

>> Please specify author contributions for C.M. and P.A. .

Done.

>> Complement your data access section with the weblinks for the data sets.

The weblinks to pdb and EMDDB entries have been added.

>> Rename the 'Methods' section to 'Material and Methods'.

Done.

>> Recheck if the Zhao et al. bioRxiv entries was published in the meantime and update the reference in case.

The preprint has not been published yet to the best of our knowledge.

>> State redisplay of Fig3B,C in the legends of FigEV5B,C.

Done.

>> Please re-format all EV files in numerical order - not skipping over a number. Update all callouts and nomenclature accordingly.

The Figures have been relabelled to Fig. EV 1-5 and callouts updated accordingly.

>>Update the Appendix ToC, adding the Appendix figure numbers.

Done.

>>Remove Movie legends from the MS. The names need correction to Movie EV#. Please zip together the corresponding movie file and movie legend in .docx format.

The movies are now called movie EV1-EV6. Extended Movies EV1-EV5 are zipped up with all movie legends under EV1-EV5 movies.zip. Note: movie EV6 has been uploaded separately because the zipped file of all six movies extended size limit of 100 MB.

>> Adjust the reference format to EMBO Journal style, limiting to 10 authors et al. .

Done.

The authors performed the requested editorial changes.

Referee #2:

I thank the authors for the substantial revision and that they sufficiently addressed the points I raised.

Three very minor things - in the methods section, one reference is missing: Baculoviruses for these constructs were generated following the standard protocol outlined in [REF].

A reference to a book chapter was added and included in the references.

In Table 1, one of the microscopes is Titan Krios, (ESRF, UK), but if it is ESRF, that is in France, unless there is another ESRF somewhere in UK, but then it is super confusing

The mistake was corrected to ESRF in France.

In Figure 2, panel H is cropped.

A corrected version of Figure 2 was uploaded to the manuscript submission system.

Overall, I recommend this manuscript for publication in the EMBO Journal

Referee #3:

The authors have addressed all of the raised minor and major points, which greatly improved the structure of the paper and makes it easier to understand. In principle, the manuscript is ready for publication. However, the authors should double-check their figures and tables which still contain some minor flaws (e.g. inconsistent formatting, missing text etc.).

Furthermore, the authors need to precisely indicate whether they are referring to wildtype or mutant proteins in their text.

Thanks for the comments. We have added missing labels and adjusted fonts in the Figures, as suggested.

Dear Dr Duerr,

Thank you for submitting the revised version of your manuscript. I have now evaluated your amended manuscript and concluded that the remaining minor concerns have been sufficiently addressed.

Thus, I am pleased to inform you that your manuscript has been accepted for publication in the EMBO Journal.

Please note that it is EMBO Journal policy for the transcript of the editorial process (containing referee reports and your response letter) to be published as an online supplement to each paper. I would thus like to ask for your consent on keeping the additional referee figures included in this file.

Also, in case you might NOT want the transparent process file published at all, you will also need to inform us via email immediately. More information is available here:
http://emboj.embopress.org/about#Transparent_Process

Please note that in order to be able to start the production process, our publisher will need and contact you regarding the following forms:

- PAGE CHARGE AUTHORISATION (For Articles and Resources)
[http://onlinelibrary.wiley.com/journal/10.1002/\(ISSN\)1460-2075/homepage/tej_apc.pdf](http://onlinelibrary.wiley.com/journal/10.1002/(ISSN)1460-2075/homepage/tej_apc.pdf)
- LICENCE TO PUBLISH (for non-Open Access)

Your article cannot be published until the publisher has received the appropriate signed license agreement. Once your article has been received by Wiley for production you will receive an email from Wiley's Author Services system, which will ask you to log in and will present them with the appropriate license for completion.

- LICENCE TO PUBLISH for OPEN ACCESS papers

Authors of accepted peer-reviewed original research articles may choose to pay a fee in order for their published article to be made freely accessible to all online immediately upon publication. The EMBO Open fee is fixed at \$5,200 (+ VAT where applicable).

We offer two licenses for Open Access papers, CC-BY and CC-BY-NC-ND.
For more information on these licenses, please visit: <http://creativecommons.org/licenses/by/3.0/> and http://creativecommons.org/licenses/by-nc-nd/3.0/deed.en_US

- PAYMENT FOR OPEN ACCESS papers

You also need to complete our payment system for Open Access articles. Please follow this link and select EMBO Journal from the drop down list and then complete the payment process:
https://authorservices.wiley.com/bauthor/onlineopen_order.asp

Should you be planning a Press Release on your article, please get in contact with embojournal@wiley.com as early as possible, in order to coordinate publication and release dates.

On a different note, I would like to alert you that EMBO Press is currently developing a new format for a video-synopsis of work published with us, which essentially is a short, author-generated film explaining the core findings in hand drawings, and, as we believe, can be very useful to increase visibility of the work. This has proven to offer a nice opportunity for exposure i.p. for the first author(s) of the study. Please see the following link for representative examples and their integration into the article web page:

<https://www.embopress.org/doi/full/10.15252/embojournal.2019103932>

If you have any questions, please do not hesitate to call or email the Editorial Office.

Kind regards,

Daniel Klimmeck

Daniel Klimmeck, PhD
Senior Editor
The EMBO Journal
EMBO
Postfach 1022-40
Meyerhofstrasse 1
D-69117 Heidelberg
contact@embojournal.org
Submit at: <http://emboj.msubmit.net>

Corresponding Author Name: Katharina L. Duerr

Journal Submitted to: EMBO J

Manuscript Number: EMBOJ-2020-107294